# Steering Large Reasoning Models towards Concise Reasoning via Flow Matching

**Yawei Li**[*]                                                                    *yawei.li@stat.uni-muenchen.de*
*LMU Munich*

**Benjamin Bergner**                                                              *bbergner@amazon.com*
*Amazon*

**Yinghan Zhao**                                                                  *yinghanz@amazon.com*
*Amazon*

**Vihang Prakash Patil**                                                          *pvihang@amazon.com*
*Amazon*

**Bei Chen**                                                                      *chenbe@amazon.com*
*Amazon*

**Cheng Wang**                                                                    *cwngam@amazon.com*
*Amazon*

**Reviewed on OpenReview:** *https://openreview.net/forum?id=qwcJMdGerK*

## Abstract

Large Reasoning Models (LRMs) excel at complex reasoning tasks, but their efficiency is often hampered by overly verbose outputs. Prior steering methods attempt to address this issue by applying a single, global vector to hidden representations—an approach grounded in the restrictive *linear representation hypothesis*. In this work, we introduce FlowSteer, a nonlinear steering method that goes beyond uniform linear shifts by learning a complete *transformation between the distributions* associated with verbose and concise reasoning. This transformation is learned via *Flow Matching* as a velocity field, enabling precise, input-dependent control over the model's reasoning process. By aligning steered representations with the distribution of concise-reasoning activations, FlowSteer yields more compact reasoning than the linear shifts. Across diverse reasoning benchmarks, FlowSteer demonstrates strong task performance and token efficiency compared to leading inference-time baselines. Our work demonstrates that modeling the full distributional transport with generative techniques offers a more effective and principled foundation for controlling LRMs.

## 1 Introduction

Recent Large Reasoning Models (LRMs), such as the OpenAI o1-series (OpenAI, 2024) and DeepSeek-R1 series (Guo et al., 2025), leverage Chain-of-Thought (CoT) reasoning (Wei et al., 2022) to tackle complex problems in domains like mathematics and coding (Ahn et al., 2024; AlphaProof & AlphaGeometry, 2024; Luo et al., 2023). By externalizing their reasoning into intermediate steps, LRMs achieve strong performance on logic-intensive tasks. However, a key challenge has emerged: their reasoning paths are often excessively verbose (Chen et al., 2025b). These over-extended traces are often filled with unnecessary self-reflection, which not only inflates computational costs but also diminishes accuracy (Chen et al., 2025a; Huang et al., 2025).

---

[*]This work was done during an internship at Amazon.

To address this inefficiency, steering methods have emerged as a promising, lightweight solution for compacting the reasoning paths of LRMs (Chen et al., 2025a; Azizi et al., 2025). These methods alter a model's behavior by directly manipulating its hidden representations at inference time. The core principle is to identify internal representations that lead to verbose outputs (source) and transform them toward representations associated with concise outputs (target). Most existing approaches, however, rely on the *linear representation hypothesis* (Park et al., 2024). This hypothesis posits that complex model behaviors can be controlled by shifting a hidden representation along a single direction, i.e. a *steering vector*. While simple to implement, this linear approach applies the same shift to all source representations, irrespective of their individual starting positions. This rigid transformation ignores the complex geometry of the underlying representation space (Wang et al., 2025b), risking pushing steered representations off the data manifold and leading to suboptimal performance (Rodriguez et al., 2025; Huang et al., 2025).

In this work, we introduce FlowSteer, a novel steering approach that does not rely on the linear representation hypothesis. Building on the perspective of steering as a distribution transport problem (Rodriguez et al., 2025), we leverage *Flow Matching* (FM) (Lipman et al., 2023; Liu, 2022) to learn a nonlinear velocity field that maps the "verbose" representation distribution to the "concise" one. This technique enables a full distributional alignment that respects the data manifold, overcoming a key limitation of prior linear approaches. However, applying Flow Matching to the LRM activation space is non-trivial; we uncover two phenomena that impede naive implementation. First, we identify that *massive activation magnitudes* severely destabilize training dynamics. We address this via a robust training strategy featuring *outlier-resistant normalization* and a *Huber* loss (Huber, 1992). Second, we observe that representations frequently stagnate in *low-velocity zones*, failing to reach the target state. To counteract this, we introduce a novel *probabilistic guidance* mechanism that ensures trajectories effectively converge to the target manifold. Importantly, FlowSteer operates while keeping the original LRM parameters intact; the resulting flow model is a lightweight MLP, adding minimal overhead during inference. Our contributions are as follows:

- We propose a novel Flow Matching-based nonlinear steering method that better preserves the target representation distribution. Quantitative experiments show that FlowSteer has roughly 5.4× better distributional alignment to target representations compared to the linear steering baseline.

- We reveal and systematically address critical challenges unique to using flow models within LRM activation space. We introduce a suite of novel techniques that ensure both stable training and the effective generation of steering trajectories, successfully overcoming pitfalls caused by massive activations and velocity stagnation.

- We demonstrate that the superior distributional alignment translates into reasoning efficiency. Specifically, we evaluate FlowSteer across five reasoning benchmarks and LRMs at 1.5B, 7B, and 32B scales. Statistical tests confirm that our approach requires fewer tokens than other inference-time baselines to achieve comparable performance. Notably, in the best case, FlowSteer achieves up to a 6.0% absolute accuracy increase *over the next-best method* while further reducing token consumption by 14.5%.

## 2 Preliminaries

### 2.1 Reducing reasoning path length by linear steering

**Hidden representation sets** Steering methods for efficient LRMs operate by intervening on the model's hidden representations $\boldsymbol{x} \in \mathbb{R}^d$ at a specific transformer block during inference. The core of these methods is the construction of two sets of representations: a source set $\mathcal{S}$ associated with *verbose* reasoning and a target set $\mathcal{T}$ associated with *concise* reasoning (Huang et al., 2025). Different approaches populate these sets in different ways. For instance, ASC (Azizi et al., 2025) and Manifold Steering (Huang et al., 2025) generate paired responses to the same prompt—a verbose CoT and a concise one—and assign representations from the former to $\mathcal{S}$ and the latter to $\mathcal{T}$. Another approach, SEAL (Chen et al., 2025a), derives these sets by analyzing a single CoT. It categorizes reasoning steps into different functional roles, identifying "Reflection/Transition" steps as verbose and "Execution" steps as concise; the hidden states corresponding to these steps are then extracted to populate $\mathcal{S}$ and $\mathcal{T}$, respectively.

**Steering vector** The steering vector $\boldsymbol{v}$ is typically defined as the difference-in-means:

$$\boldsymbol{v} = \frac{1}{|\mathcal{T}|} \sum_{\boldsymbol{x} \in \mathcal{T}} \boldsymbol{x} - \frac{1}{|\mathcal{S}|} \sum_{\boldsymbol{x} \in \mathcal{S}} \boldsymbol{x}, \tag{1}$$

where $\boldsymbol{v}$ encodes the direction toward more concise reasoning paths. To enhance robustness against noise in the hidden representations, Manifold Steering (Huang et al., 2025) additionally applies a Principal Component Analysis (PCA) projection to $\boldsymbol{v}$. At inference time, once a token is identified for intervention, its representation $\boldsymbol{x}$ is shifted using the steering vector:

$$\boldsymbol{x}' = \boldsymbol{x} + \gamma \boldsymbol{v}, \tag{2}$$

where $\boldsymbol{x}'$ is the updated representation. The coefficient $\gamma \in \mathbb{R}$ is either a fixed hyperparameter (Chen et al., 2025a; Zhao et al., 2025; Azizi et al., 2025) or a value dependent on $\boldsymbol{x}$ (Huang et al., 2025). Because these methods adjust hidden representations via a simple affine transformation, we refer to them as *linear steering methods* and term the coefficient $\gamma$ the *linear strength*.

## 2.2 Preliminaries on Flow Matching

**From interpolation to velocity fields** Flow Matching is a recent generative modeling technique to learn a smooth interpolation between a source distribution $p_0$ and a target distribution $p_1$, enabling sampling from $p_1$ given samples from $p_0$. Given a sample pair from the joint distribution $(\boldsymbol{x}_0, \boldsymbol{x}_1) \sim \pi_{0,1}$, a linear path can be constructed as: $\boldsymbol{x}_t = t\boldsymbol{x}_1 + (1-t)\boldsymbol{x}_0$ for $t \in [0, 1]$, which specifies the position at time $t$. Flow Matching characterizes the motion of $\boldsymbol{x}_t$, which later serves as the basis for generating new samples. This motion is described mathematically by a time-dependent velocity field $u : \mathbb{R}^d \times [0, 1] \to \mathbb{R}^d$. Taking the time derivative of $\boldsymbol{x}_t$ yields the *conditional velocity field*: $u_t(\boldsymbol{x}_t | \boldsymbol{x}_1, \boldsymbol{x}_0) = \frac{\mathrm{d}\boldsymbol{x}_t}{\mathrm{d}t} = \boldsymbol{x}_1 - \boldsymbol{x}_0$. However, $\boldsymbol{x}_1$ is unavailable when generating new samples. To obtain a velocity field that depends only on $(\boldsymbol{x}_t, t)$, Flow Matching learns to model the *marginal velocity field*:

$$u_t(\boldsymbol{x}_t) = \mathbb{E}_{p_t(\boldsymbol{x}_1, \boldsymbol{x}_0 | \boldsymbol{x}_t)} \left[ u_t(\boldsymbol{x}_t | \boldsymbol{x}_1, \boldsymbol{x}_0) \right]. \tag{3}$$

**Conditional Flow Matching loss** The marginal velocity field can be learned by minimizing the regression loss $\mathcal{L}_{\mathrm{FM}}(\boldsymbol{\theta}) = \mathbb{E}_{t, p_t(\boldsymbol{x}_t)}[||v_{\boldsymbol{\theta}}(\boldsymbol{x}_t, t) - u_t(\boldsymbol{x}_t)||_2^2]$, where $v_{\boldsymbol{\theta}}$ is the *flow model* parametrized by $\boldsymbol{\theta}$. However, computing $u_t(\boldsymbol{x}_t)$ requires marginalization over all $(\boldsymbol{x}_0, \boldsymbol{x}_1)$, making $\mathcal{L}_{\mathrm{FM}}(\boldsymbol{\theta})$ intractable. To address this issue, Flow Matching introduces the *conditional Flow Matching loss*:

$$\mathcal{L}_{\mathrm{CFM}}(\boldsymbol{\theta}) = \mathbb{E}_{t \sim \mathcal{U}[0,1], (\boldsymbol{x}_0, \boldsymbol{x}_1) \sim \pi_{0,1}} \left[ ||v_{\boldsymbol{\theta}}(\boldsymbol{x}_t, t) - u_t(\boldsymbol{x}_t | \boldsymbol{x}_1, \boldsymbol{x}_0)||_2^2 \right]. \tag{4}$$

Lipman et al. (2023); Liu (2022) show that $\mathcal{L}_{\mathrm{CFM}}(\boldsymbol{\theta})$ and $\mathcal{L}_{\mathrm{FM}}(\boldsymbol{\theta})$ yield identical gradients, so optimizing $\mathcal{L}_{\mathrm{CFM}}(\boldsymbol{\theta})$ effectively learns the marginal velocity field $u_t(\boldsymbol{x}_t)$.

**New sample generation** Once the flow model $v_{\boldsymbol{\theta}}(\boldsymbol{x}_t, t)$ is trained, new samples can be generated by using it as an approximation of the marginal velocity field $u_t(\boldsymbol{x}_t)$. Starting with an initial sample $\boldsymbol{x}_0 \sim p_0$, one simulates the Ordinary Differential Equation (ODE) defined by the learned model: $\frac{\mathrm{d}\boldsymbol{x}_t}{\mathrm{d}t} = v_{\boldsymbol{\theta}}(\boldsymbol{x}_t, t)$, from $t = 0$ to $t = 1$. In practice, the ODE is solved numerically using methods such as Euler or Dopri5 (Dormand & Prince, 1980). Two important properties of Flow Matching are worth highlighting: (1) By construction, the marginal velocity field $u_t(\boldsymbol{x}_t)$ transports the source distribution $p_0$ exactly to the target distribution $p_1$. Consequently, if the flow model $v_{\boldsymbol{\theta}}(\boldsymbol{x}_t, t)$ perfectly recovers $u_t(\boldsymbol{x}_t)$ and is used for generation, the resulting samples also follow $p_1$. (2) Although the conditional velocity field is derived from a linear interpolation between $\boldsymbol{x}_0$ and $\boldsymbol{x}_1$, the trajectories produced by the marginal velocity field during generation are generally curved. These properties are central to our motivation for grounding method in Flow Matching, which we introduce next.

## 3 Methodology

**Distribution alignment via Flow Matching** As introduced earlier, hidden states linked to verbose CoTs form the source distribution, while those linked to concise CoTs form the target distribution (Figure 1,

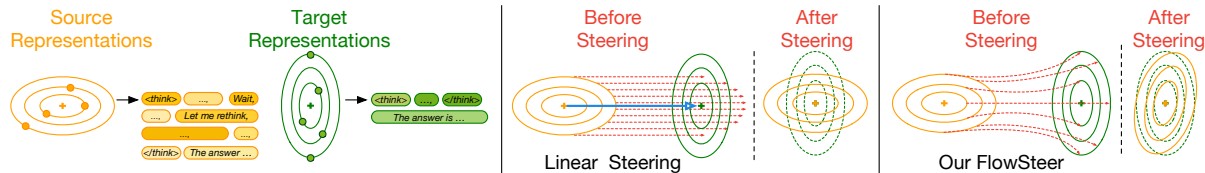

Figure 1: *Left*: The **source** distribution corresponds to hidden representations that produce **verbose** CoTs, while the **target** distribution corresponds to representations that produce **concise** CoTs. Better zoom in for clarity. *Middle*: Linear steering methods apply the same steering vector (the blue bolded arrow) to all source representations, aligning only the means of the two distributions. This ignores higher-order statistics such as covariance, often resulting in a substantial mismatch. *Right*: Our FlowSteer leverages Flow Matching to learn a mapping from the source distribution to the target distribution, naturally aligning the two due to the theoretical properties of Flow Matching.

left). The goal of steering is to *transport the source distribution to the target* (Rodriguez et al., 2025), guiding the LRM to produce concise CoTs. Constrained by the linear representation hypothesis, existing steering methods simplify this transport into a uniform shift. This approach, however, only aligns the means of the two distributions while ignoring higher-order statistics, often resulting in a distributional mismatch (Figure 1, middle). In contrast, our FlowSteer is grounded in Flow Matching, whose marginal velocity field guarantees an exact transport between arbitrary distributions (Figure 1, right). This capability align steered representations with the target geometry, minimizing off-manifold excursions that trigger degenerative repetition or reasoning collapse (Huang et al., 2025). Furthermore, the ability to induce *nonlinear steering* trajectories marks a key distinction from linear steering (Chen et al., 2025a; Huang et al., 2025; Rodriguez et al., 2025).

**Challenges of flowing in LRM activation space** Despite the appealing properties of Flow Matching for distribution transport, applying it in the hidden space of LRMs introduces significant challenges. (1) The first arises from *massive activations* (Sun et al., 2024): a subset of activations consistently exhibits magnitudes orders larger than the rest, often dominating the global norm of the hidden states. These extreme values can severely destabilize training by inflating the MSE loss. This prevents the flow model from learning useful structure and leads to divergent generation trajectories, pushing the steered representation toward infinity. (2) The second challenge is the emergence of *low-velocity zones*: regions between the source and target manifolds where the learned velocity field has a very small magnitude. If a representation $\boldsymbol{x}_0$ lies in such a zone, it may move only minimally toward the target distribution and lead to suboptimal performance. Next, we elaborate on these challenges and our solutions.

## 3.1 Robust training strategy

**Massive activations cause divergent steering trajectories** In typical application domains of Flow Matching, e.g. image generation, the source $\boldsymbol{x}_0$ is standard Gaussian noise and the target pixels $\boldsymbol{x}_1$ are bounded in $[0, 1]$. Consequently, the regression target $u_t(\boldsymbol{x}_t|\boldsymbol{x}_1, \boldsymbol{x}_0) = \boldsymbol{x}_1 - \boldsymbol{x}_0$ in Eq. (4) typically has a moderate scale and limited variance. In contrast, activations in LRMs are unbounded and can contain substantial outliers (Sun et al., 2024), as illustrated for the first four dimensions of the representations in Figure 2. These massive activations inflate the variance of the loss and destabilize training. The flow model then disproportionately allocates capacity to predicting conditional velocities associated with these extremes, underfitting the typical structure of the source and target distributions. At steering time, the poorly fitted flow model makes large prediction errors in certain regions, pushing representations off the target manifold and causing steering trajectories to diverge toward infinity. To mitigate this problem, we explicitly reduce the influence of outlier activations by introducing robust data normalization and a robust loss.

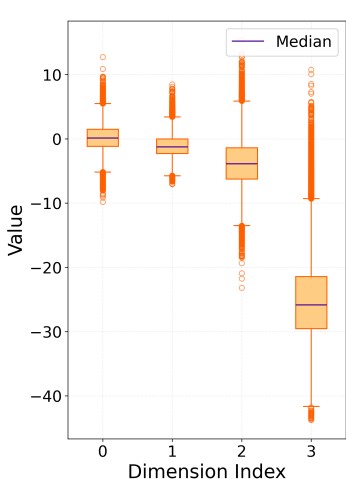

Figure 2: Activations from layer 20 of DeepSeek-R1-Distill-Qwen-1.5B.

**Robust data normalization**   The presence of massive activations necessitates a robust normalization of the source and target representations. Conventional dimensionwise z-score scaling ($\frac{x-\mu}{\sigma}$) is unsuitable for this landscape, as outliers may skew the estimated mean $\mu$ and inflate the standard deviation $\sigma$. This effect compresses the majority of typical activations into a narrow range, masking important structural information. Indeed, our ablation study in Section 4.4 confirms that this information loss is fatal: training the flow model with z-score normalization leads to model collapse, yielding near-zero accuracy across multiple benchmarks. We therefore adopt a more robust normalization using the median and interquartile range (*median-IQR normalization*), which we apply independently to the source and target representation sets. For a given activation value $x$ in a single dimension (of $\boldsymbol{x}_0$ or $\boldsymbol{x}_1$), its normalized value $\tilde{x}$ is computed as

$$\tilde{x} = \frac{x - \text{median}(X)}{Q_{75}(X) - Q_{25}(X)}, \tag{5}$$

where $\text{median}(X)$, the 75th percentile $Q_{75}(X)$, and the 25th percentile $Q_{25}(X)$ are computed for that dimension across the training set (of either source or target representations). This method is inherently resilient to outliers, preserving meaningful variations within the non-outlier data.

**Robust loss function**   The standard MSE loss $||v_{\boldsymbol{\theta}}(\boldsymbol{x}_t, t) - (\boldsymbol{x}_1 - \boldsymbol{x}_0)||_2^2$ in the conditional Flow Matching loss (Eq. (4)) is also highly sensitive to outliers. To improve training stability, we replace it with a more robust, dimension-wise *Huber loss* (Huber, 1992). Let $\boldsymbol{\zeta} = v_{\boldsymbol{\theta}}(\boldsymbol{x}_t, t) - (\boldsymbol{x}_1 - \boldsymbol{x}_0)$ be the prediction residual vector. The Huber loss for its $k$-th dimension is:

$$D_{\text{Huber}}(\zeta_k) = \begin{cases} \frac{1}{2}\zeta_k^2 & \text{if } |\zeta_k| \leq 1, \\ |\zeta_k| - \frac{1}{2} & \text{otherwise.} \end{cases} \tag{6}$$

The total loss is obtained by averaging across all $d$ dimensions. This loss replaces the MSE loss in Eq. (4), and the final optimization objective is the expectation of this total Huber loss over time $t \sim \mathcal{U}[0,1]$ and data pairs $(\boldsymbol{x}_0, \boldsymbol{x}_1) \sim \pi_{0,1}$. The Huber loss behaves quadratically for small residuals ($|\zeta_k| \leq 1$) and linearly for large ones, reducing the influence of outliers. As training progresses, the residual values $\zeta_k$ typically converge to be smaller than 1. In this regime, the Huber loss becomes equivalent to the squared loss, thereby preserving the key theoretical guarantee of Flow Matching such as the relationship $\nabla \mathcal{L}_{\text{CFM}}(\boldsymbol{\theta}) = \nabla \mathcal{L}_{\text{FM}}(\boldsymbol{\theta})$, as introduced in Section 2.2.

**Better source-target coupling**   The standard approach in Eq. (4) couples source $\boldsymbol{x}_0$ and target $\boldsymbol{x}_1$ samples independently (i.e., $\pi_{0,1}(\boldsymbol{x}_0, \boldsymbol{x}_1) = p_0(\boldsymbol{x}_0)p_1(\boldsymbol{x}_1)$), which can create unnecessarily complex paths for the flow model to learn. To address this, we adopt the minibatch 2-Wasserstein Optimal Transport (OT) coupling strategy (Tong et al., 2024). This strategy finds an optimal pairing of source and target samples within each batch, which simplifies the resulting conditional velocity fields and thus makes the learning objective easier to optimize. In practice, the OT plan $\pi_{0,1}^{\text{OT}}$ is computed efficiently using a solver such as the Sinkhorn algorithm (Cuturi, 2013).

The robust training techniques above successfully resolve the issue of divergent steering trajectories, eliminating infinite values in steered representations. The second challenge, the existence of low-velocity zones, becomes the primary obstacle to effective steering. In the following section, we analyze the underlying causes of these zones and introduce our probabilistic guidance as the solution.

## 3.2   Probabilistic guidance avoids stagnation in low-velocity zones

**Causes of low-velocity zones**   We refer a low-velocity zone to a region where the velocity vectors of the learned field have a small magnitude. The formation of these zones stems from the conditional Flow Matching loss, $\mathcal{L}_{\text{CFM}}(\boldsymbol{\theta})$ (Eq. (4)), where the ground-truth regression target is the conditional velocity $u_t(\boldsymbol{x}_t|\boldsymbol{x}_1, \boldsymbol{x}_0) = \boldsymbol{x}_1 - \boldsymbol{x}_0$. Prior studies (Chen et al., 2025a; Huang et al., 2025) indicate that the source ($p_0$) and target ($p_1$) distributions maintain significant spatial proximity in the representation space. Consequently, training pairs $(\boldsymbol{x}_0, \boldsymbol{x}_1)$ sampled from these proximal regions naturally yield velocity targets with small magnitudes ($\boldsymbol{x}_1 - \boldsymbol{x}_0 \approx 0$). Furthermore, the minibatch OT plan exacerbates the formation of low-velocity zones. By design, OT minimizes transport cost by coupling pairs $(\boldsymbol{x}_0, \boldsymbol{x}_1)$ that are already close. This strategy systematically pairs more points within the proximal regions, which in turn generates more small-magnitude

regression targets and encourages the flow model to learn a low-velocity field there. Consequently, at steering time, any source representation $\boldsymbol{x}_0$ that starts in this zone will stagnate, leading to suboptimal compression of the reasoning paths.

**Probabilistic guidance** Inspired by guidance techniques in conditional image generation (Dhariwal & Nichol, 2021; Ho & Salimans, 2021; Karras et al., 2024), we introduce a *probabilistic guidance* to help representations escape low-velocity zones. Unlike in image generation, where guidance aims to align outputs with text prompts, our goal is to steer representations from the source distribution $p_0$ toward the target distribution $p_1$. Intuitively, this process should increase the likelihood $p_1(\boldsymbol{x}_t)$ while decreasing $p_0(\boldsymbol{x}_t)$ along the trajectory. A standard approach is *classifier guidance* (Dhariwal & Nichol, 2021), which derives a guidance from the score function $\nabla_{\boldsymbol{x}} \log q(c = 1|\boldsymbol{x}_t)$ of an auxiliary classifier trained to distinguish between the source (class $c = 0$) and target (class $c = 1$) distributions. However, we avoid this method due to the computational overhead of training and deploying an additional classifier.

Instead, we propose a *training-free* guidance mechanism that eliminates the need for additional learnable parameters. Our approach is motivated by prior work (Hashemi et al., 2021; Zhang et al., 2021) showing that activation distributions in the later layers of neural networks are well-approximated by Gaussians, an approximation also adopted in previous LLM steering research (Rodriguez et al., 2025). Since we perform steering in later LRM layers (see Section A), we approximate both the source distribution $p_0$ and the target distribution $p_1$ with Gaussians: $p_0(\cdot) \approx \mathcal{N}(\cdot; \boldsymbol{\mu}_0, \boldsymbol{\Sigma}_0)$ and $p_1(\cdot) \approx \mathcal{N}(\cdot; \boldsymbol{\mu}_1, \boldsymbol{\Sigma}_1)$. This allows us to analytically compute the difference between their score functions:

$$g_t(\boldsymbol{x}_t) = \nabla_{\boldsymbol{x}} \log p_1(\boldsymbol{x}_t) - \nabla_{\boldsymbol{x}} \log p_0(\boldsymbol{x}_t) = \boldsymbol{\Sigma}_1^{-1}(\boldsymbol{\mu}_1 - \boldsymbol{x}_t) - \boldsymbol{\Sigma}_0^{-1}(\boldsymbol{\mu}_0 - \boldsymbol{x}_t). \tag{7}$$

This vector $g_t(\boldsymbol{x}_t)$ points in the direction that maximally increases the log-likelihood of the target distribution while decreasing that of the source. Let $\bar{g}_t(\boldsymbol{x}_t) = g_t(\boldsymbol{x}_t)/\|g_t(\boldsymbol{x}_t)\|_2$ be its normalized version. We incorporate this guidance into the generation ODE $\frac{\mathrm{d}\boldsymbol{x}_t}{\mathrm{d}t} = v_{\boldsymbol{\theta}}(\boldsymbol{x}_t, t)$ as follows:

$$\frac{\mathrm{d}\boldsymbol{x}_t}{\mathrm{d}t} = v_{\boldsymbol{\theta}}(\boldsymbol{x}_t, t) + \eta g_t(\boldsymbol{x}_t) - \left(v_{\boldsymbol{\theta}}(\boldsymbol{x}_t, t)^\top \bar{g}_t(\boldsymbol{x}_t)\right) \bar{g}_t(\boldsymbol{x}_t), \tag{8}$$

where $\eta \in \mathbb{R}_+$ is a hyperparameter for guidance strength. The final term, $(v_{\boldsymbol{\theta}}(\boldsymbol{x}_t, t)^\top \bar{g}_t(\boldsymbol{x}_t))\bar{g}_t(\boldsymbol{x}_t)$, is the projection of the learned velocity $v_{\boldsymbol{\theta}}(\boldsymbol{x}_t, t)$ onto the guidance direction $\bar{g}_t(\boldsymbol{x}_t)$. This formulation allows the guidance to operate *adaptively* depending on the representation's location:

(1) When $\boldsymbol{x}_t$ is in a low-velocity zone, the learned velocity $v_{\boldsymbol{\theta}}(\boldsymbol{x}_t, t)$ has small magnitude. Consequently, its projection is negligible, and the dynamics are dominated by the guidance term $\eta g_t(\boldsymbol{x}_t)$. As defined in Eq. (7), this guidance vector is non-zero almost everywhere as long as the source and target distributions are distinct (i.e., have different means or covariances). It thus provides a persistent force throughout the low-velocity zones, pushing the representation toward the target and effectively helping it escape stagnation.

(2) When $\boldsymbol{x}_t$ is outside a low-velocity zone, the magnitude of $v_{\boldsymbol{\theta}}(\boldsymbol{x}_t, t)$ is larger. Both $v_{\boldsymbol{\theta}}(\boldsymbol{x}_t, t)$ and $\bar{g}_t(\boldsymbol{x}_t)$ generally point toward the target region, making their inner product positive. In this regime, the projection term subtracts a portion of the guidance, reducing its influence. This allows the trajectory to be primarily determined by the more nuanced, learned flow model $v_{\boldsymbol{\theta}}(\boldsymbol{x}_t, t)$, which captures the finer-grained structure of the target manifold.

**Practical considerations** Computing the matrix inverse $\boldsymbol{\Sigma}^{-1}$ in Eq. (7) is computationally challenging in high-dimensional spaces. To reduce this overhead, we employ a diagonal factorization for $\boldsymbol{\Sigma}$, a technique consistent with recent research on LLMs that also handles high-dimensional Gaussians (Yang et al., 2024; Li et al., 2025; Rodriguez et al., 2025). Specifically, we set $\boldsymbol{\Sigma}_1 = \mathrm{diag}(\boldsymbol{\sigma}_1 \odot \boldsymbol{\sigma}_1)$ and $\boldsymbol{\Sigma}_0 = \mathrm{diag}(\boldsymbol{\sigma}_0 \odot \boldsymbol{\sigma}_0)$, where $\odot$ denotes the Hadamard product. The vectors $\boldsymbol{\sigma}_1, \boldsymbol{\sigma}_0 \in \mathbb{R}_+^d$ contain the per-dimension standard deviations, which we estimate from the training data along with the means $\boldsymbol{\mu}_0$ and $\boldsymbol{\mu}_1$. While this factorization is less expressive than the full covariance structure, our experiments in Sections 4.4 and 4.5 show that the guidance substantially boosts accuracy and token efficiency, justifying the factorization choice.

# 4 Experiments

## 4.1 Implementation

**Steering layer and steering tokens**  Representation steering involves two key components: a *protocol* that specifies where to intervene (i.e., at which layers and tokens) and a mechanism that defines how to modify the representations. Focusing on the latter, our primary contribution is FlowSteer, a steering mechanism. To isolate the impact of our approach, we conduct a controlled comparison against one of the state-of-the-art linear methods, SEAL (Chen et al., 2025a). We adopt SEAL's intervention protocol by steering the same layer and intervening at every "\n\n" token, which serves as the delimiter for splitting reasoning steps. By retaining this protocol while replacing the linear mechanism with FlowSteer, we ensure that any observed improvements are attributable solely to our proposed steering mechanism. Further implementation details are available in Section A.

**Training dataset**  To ensure a fair comparison and equivalent data efficiency, we use an *identical* training dataset for both SEAL and FlowSteer. We extract representations from the **MATH/train** set (Hendrycks et al., 2021). These representations are then used for two purposes: (1) to compute the steering vector for SEAL, and (2) to train our MLP flow model, which includes the estimation of distributional statistics (e.g., percentiles, mean). Notably, the number of question samples used for representation extraction is modest, ranging from 1,000 to 3,600. Even for a 32B LRM, our flow model can be trained on a single GPU within 24 hours. Consequently, compared to RL-based approaches (e.g., Dai et al. (2025); Xiang et al. (2025)), our method is significantly less demanding in terms of data samples and computational resources. We outline the specific training details in Table 5.

**Evaluation benchmarks**  For math, we use: **MATH500** (Lightman et al., 2024; Hendrycks et al., 2021), **GSM8K** (Cobbe et al., 2021), **AIME24** (Mathematical Association of America, 2024), and **AMC23** (Mathematical Association of America, 2023). To test the cross-domain generalization, we evaluate on the coding task **LiveCodeBench** (Jain et al., 2025).

**LRMs**  To evaluate the generalizability of our approach across different model families and scales, we conduct experiments on Deepseek-R1-distill-Qwen-1.5B (Guo et al., 2025) and its 7B variant (denoted with **R1-1.5B/7B**), and Qwen-QwQ-32B (Qwen-Team, 2025) (denoted with **QwQ-32B**). To maintain a controlled environment, we follow Chen et al. (2025a) and use greedy decoding for the vanilla models and all inference-time approaches (listed in Section 4.3). We set the maximal token length of LRMs to 15,000 on AIME24 and 10,000 on all other benchmarks.

**Flow model and guidance strength**  Our flow model $v_{\boldsymbol{\theta}}$ is a lightweight MLP, adding minimal computational overhead, as detailed in Section 4.5. Specifically, we use a 6-layer MLP for the R1-1.5B/7B models and an 8-layer MLP for the QwQ-32B model. During steering, we generate trajectories using the Dopri5 ODE solver. To ensure a fair comparison of performance, we conduct a hyperparameter sweep for both the guidance strength $\eta$ of FlowSteer and the linear strength $\gamma$ of SEAL, reporting the best-performing configuration for each. The detailed hyperparameter values are reported in Section A.4. We note that a default $\eta = 1$ is a strong choice across most models and datasets.

**Baseline methods**  We compare FlowSteer against a diverse set of inference-time intervention methods. These include the linear steering method **SEAL** (Chen et al., 2025a), a logit-level intervention method **LogitsPenalty** (Wang et al., 2025c), two token-level methods, **AlphaOne** (Zhang et al., 2025) and **s1\*** (Muennighoff et al., 2025), and the prompt-based method **CoD** (Xu et al., 2025a). The s1\* is the version without supervised fine-tuning. In addition, we report results from a single run, which aligns with the standard evaluation settings established by previous works (Chen et al., 2025a; Zhang et al., 2025; Wang et al., 2025c). In line with SEAL and AlphaOne, we do not compare against RL-based approaches. These methods represent an orthogonal line of work focused on optimizing LRM weights to improve CoT generation. In contrast, the methods evaluated here are inference-time interventions that do not alter the base LRM. Notably, these two approaches are not mutually exclusive; inference-time approaches could potentially be applied to RL-optimized LRMs to further control the reasoning paths.

Table 1: Distributional distance between the source/steered representations and the target ones. FlowSteer achieves a substantially better alignment (lower distance) to the target than both the original source representations ("Before Steering") and those steered by SEAL.

| Methods | R1-1.5B | | | R1-7B | | | QwQ-32B | | |
|---|---|---|---|---|---|---|---|---|---|
| | MMD ↓ | FID ↓ | KID ↓ | MMD ↓ | FID ↓ | KID ↓ | MMD ↓ | FID ↓ | KID ↓ |
| Before Steering | 564.6 | 8227.4 | 1137.5 | 577.1 | 22784.8 | 1609.5 | 652.9 | 67647.1 | 33951.7 |
| After SEAL | 386.2 | 4639.6 | 69.8 | 517.0 | 17158.1 | 189.4 | 539.9 | 44987.7 | 766.6 |
| After **FlowSteer** | **98.0** | **690.2** | **11.4** | **105.3** | **4701.4** | **25.3** | **60.4** | **5658.2** | **293.1** |

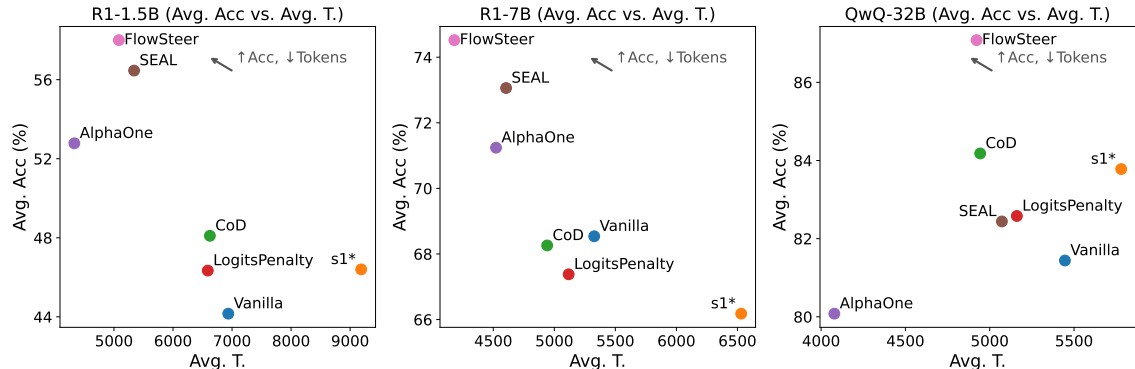

Figure 3: Average accuracy and token count aggregated across all benchmarks.

## 4.2 Alignment between steered and target distributions

**FlowSteer achieves superior distributional alignment**   To quantify how well steered representations align with the target ones, we measure their distributional distance before and after steering. Both source and target representations are extracted from the MATH/train set. We first establish a baseline distance between the original source and target distributions ("Before Steering"). Next, we apply SEAL and FlowSteer to the source representations and measure the distance from each of the two resulting steered distributions to the target. We use three metrics: Maximum Mean Discrepancy (*MMD*) (Gretton et al., 2012), Fréchet Inception Distance (*FID*) (Heusel et al., 2017), and Kernel Inception Distance (*KID*) (Sutherland et al., 2018) (detailed in Section E). As shown in Table 1, FlowSteer consistently reduces distributional distance by a large margin across all LRMs, often by more than an order of magnitude compared to the unsteered baseline. Moreover, it outperforms the linear steering method SEAL, achieving distances that are **5.4× lower** in average. These results demonstrate that FlowSteer produces representations better aligned with the target manifold than linear steering, providing a strong quantitative explanation for its task performance reported in Table 2, which we will analyze in the next subsection.

## 4.3 Evaluation on mathematical and coding tasks

**FlowSteer achieves a competitive accuracy–token count trade-off**   As summarized by the average columns in Table 2 and the scatter plots in Figure 3, FlowSteer consistently lies on the empirical accuracy–token Pareto frontier across all three LRMs. Specifically, FlowSteer achieves the best of both worlds on R1-7B with the highest accuracy and lowest token count; meanwhile, on R1-1.5B and QwQ-32B, while there are baselines with lower token usage than FlowSteer, they substantially sacrifice accuracy. That said, there is an inherent trade-off between accuracy and token usage: deeper reasoning can improve accuracy, but may consume more tokens. On challenging math benchmarks like AIME24 and AMC23, the accuracy gains of FlowSteer are particularly striking. For instance, with R1-1.5B on AMC23, FlowSteer reaches 72.5% accuracy, a **6.0% absolute improvement** over the strongest baseline (66.5%) while further reducing the token count by 14.5%. Furthermore, this strong performance is not limited to math; FlowSteer achieves the highest accuracy across all three LRMs on the LiveCodeBench coding task, demonstrating its cross-domain

Table 2: Performance across various math and coding benchmarks. "**Acc.**" (↑) denotes Pass@1 accuracy (%) . "**T.**" (↓) is the average number of generated tokens across *all answers*, and "**T.@C**" (↓) is the average number of tokens in *correct answers*. The best and second-best values are highlighted with **bold** and underline, respectively. "Vanilla" refers to the vanilla LRM without intervention. "Average" shows the average performance across all benchmarks.

| Methods | MATH500 | | | GSM8K | | | AIME24 | | | AMC23 | | | LiveCodeBench | | | Average | | |
|---|---|---|---|---|---|---|---|---|---|---|---|---|---|---|---|---|---|---|
| | Acc. | T. | T.@C | Acc. | T. | T.@C | Acc. | T. | T.@C | Acc. | T. | T.@C | Acc. | T. | T.@C | Acc. | T. | T.@C |
| *DeepSeek-R1-Distill-Qwen-1.5B* | | | | | | | | | | | | | | | | | | |
| Vanilla | 66.6 | 4785 | 2390 | 73.8 | 2072 | 819 | 6.7 | 13807 | **2735** | 52.5 | 6269 | 2998 | 21.2 | 7749 | 2372 | 44.2 | 6936 | **2263** |
| s1* | 69.8 | 7428 | 6188 | 69.0 | 6849 | 5799 | 26.7 | 14334 | 12498 | 47.5 | 9020 | 7936 | 19.0 | 8308 | 2606 | 46.4 | 9188 | 7005 |
| CoD | 73.0 | 4040 | 2093 | 79.5 | 2044 | 1004 | 20.0 | 12419 | 4264 | 47.5 | 6561 | 3000 | 20.5 | 8061 | 2394 | 48.1 | 6625 | 2551 |
| LogitsPenalty | 72.6 | 3847 | 1988 | 77.9 | 1360 | 715 | 10.0 | 13538 | 8801 | 50.0 | 6454 | 2709 | 21.2 | 7749 | 2331 | 46.3 | 6590 | 3309 |
| AlphaOne | 75.6 | 3858 | 3472 | 78.2 | 996 | 946 | 23.3 | **7314** | 6304 | 62.5 | 4550 | 4060 | 24.3 | **4927** | 4040 | 52.8 | **4329** | 3764 |
| SEAL | 78.2 | 3194 | 2003 | **81.0** | 968 | 732 | **33.3** | 10517 | 4792 | 66.5 | 4810 | 3256 | 23.3 | 7217 | 2380 | 56.5 | 5341 | 2633 |
| **FlowSteer** | **79.6** | **2915** | **1948** | 80.1 | **961** | **700** | **33.3** | 10211 | 3884 | **72.5** | **4111** | **2565** | **24.5** | 7217 | **2327** | **58.0** | 5083 | 2285 |
| *DeepSeek-R1-Distill-Qwen-7B* | | | | | | | | | | | | | | | | | | |
| Vanilla | 87.0 | 3341 | 2605 | 87.9 | 1160 | 867 | 50.0 | 9903 | 5539 | 72.5 | 5431 | 3698 | 45.3 | 6799 | 3394 | 68.5 | 5327 | 3221 |
| s1* | 84.0 | 4846 | 4013 | 88.3 | 2782 | 2181 | 43.3 | 12060 | 8214 | 70.0 | 5990 | 4270 | 45.3 | 6976 | 3652 | 66.2 | 6531 | 4466 |
| CoD | 89.0 | 2397 | 2143 | 87.5 | 925 | 867 | 40.0 | 10595 | 4810 | 80.0 | 4360 | 2999 | 44.8 | 6431 | 3331 | 68.3 | 4942 | 2830 |
| LogitsPenalty | 88.0 | 2978 | 2333 | 87.4 | 902 | 758 | 40.0 | 10457 | 4914 | 77.5 | 4764 | 3244 | 44.0 | 6487 | 3367 | 67.4 | 5118 | 2923 |
| AlphaOne | 87.4 | 4070 | 3797 | **90.1** | 862 | 851 | 46.7 | **7605** | 6052 | 82.5 | 4820 | 4221 | 49.5 | **5256** | 4769 | 71.2 | 4523 | 3938 |
| SEAL | **90.2** | 2613 | 2158 | 88.4 | 846 | 754 | 46.7 | 9934 | 5223 | **90.0** | 3627 | 2919 | 50.0 | 6005 | **3183** | 73.1 | 4605 | 2847 |
| **FlowSteer** | **90.2** | **2549** | **2010** | 88.6 | **797** | **732** | **53.3** | 8453 | 4132 | **90.0** | **3177** | **2739** | **50.5** | 5930 | 3204 | **74.5** | **4181** | **2563** |
| *Qwen-QwQ-32B* | | | | | | | | | | | | | | | | | | |
| Vanilla | 90.8 | 3549 | 3027 | 95.6 | 1157 | 1069 | 63.3 | 10791 | 8486 | 80.0 | 6053 | 5067 | 77.5 | 5680 | 4494 | 81.4 | 5446 | 4429 |
| s1* | 91.4 | 4180 | 3744 | 96.3 | 1858 | 1753 | 70.0 | 10852 | 9073 | 85.0 | 6275 | 5618 | 76.2 | 5734 | 4479 | 83.8 | 5780 | 4933 |
| CoD | 91.0 | 3672 | 3346 | 96.1 | 1008 | 971 | 70.0 | 9498 | 8976 | 85.0 | 5169 | 4316 | 78.8 | 5370 | 4191 | 84.2 | 4943 | 4360 |
| LogitsPenalty | 90.2 | 3135 | 2746 | 95.6 | 956 | 906 | 63.3 | 9942 | 8831 | 85.0 | 5779 | 5381 | 78.8 | 5993 | 4070 | 82.6 | 5161 | 4387 |
| AlphaOne | 91.6 | **3067** | 2908 | 94.8 | **866** | 876 | 53.3 | **6544** | 5165 | 82.5 | **4082** | **3573** | 78.2 | 5834 | 5646 | 80.1 | **4079** | **3634** |
| SEAL | **92.8** | 3102 | 2732 | 96.4 | 976 | 905 | 60.0 | 10795 | 8665 | 82.5 | 5452 | 4487 | 80.5 | 5036 | 3981 | 82.4 | 5072 | 4154 |
| **FlowSteer** | 91.0 | 3118 | **2675** | **96.5** | 917 | 857 | **76.7** | 9848 | 8355 | **90.0** | 5698 | 5220 | **81.2** | **5030** | 3970 | **87.1** | 4922 | 4215 |

Figure 4: Average accuracy as a function of average token count for correct answers.

versatility. To evaluate token efficiency, we conduct a Wilcoxon signed-rank test in Section B, comparing the average token counts of FlowSteer against the most accurate baseline. We restrict this comparison to benchmarks where FlowSteer and the strongest baseline attain comparable accuracy (an absolute difference below 5 percentage points). We introduce this constraint because a direct comparison of token counts can be misleading, as attaining higher accuracy may require longer CoTs, a relationship we elaborate on in the

Table 3: Ablation study on *R1-1.5B*. Results show that: (1) Median-IQR normalization and Huber loss are both essential for flow model training; their omission causes the steered LRM to produce random outputs until the maximum token limit is exhausted. (2) Probabilistic guidance provides significant gains in accuracy and token efficiency. See Section D.1 for additional LRM results.

| Data normalization + Loss | MATH500 | | | GSM8K | | | AIME24 | | | AMC23 | | | LiveCodeBench | | |
|---|---|---|---|---|---|---|---|---|---|---|---|---|---|---|---|
| | Acc. | T. | T.@C | Acc. | T. | T.@C | Acc. | T. | T.@C | Acc. | T. | T.@C | Acc. | T. | T.@C |
| Vanilla LRM | 66.6 | 4785 | 2390 | 73.8 | 2072 | 819 | 6.7 | 13807 | **2735** | 52.5 | 6269 | 2998 | 21.2 | 7749 | 2372 |
| Z-score + MSE | 0.4 | 9980 | 2815 | 0.8 | 9421 | 2111 | 0.0 | 14999 | - | 0.0 | 10000 | - | 0.3 | 9986 | 4715 |
| Z-score + Huber | 0.8 | 9799 | 2865 | 0.5 | 9980 | 2804 | 0.0 | 15000 | - | 0.0 | 10000 | - | 0.5 | 9676 | 2851 |
| Median-IQR + MSE | 1.7 | 9735 | 2666 | 1.0 | 9557 | 1411 | 0.0 | 15000 | - | 0.0 | 10000 | - | 1.0 | 9519 | 2754 |
| Median-IQR + Huber | 78.6 | 3168 | 2026 | **81.2** | 1071 | 782 | 13.3 | 11534 | 2996 | 67.5 | 4688 | 2919 | 21.0 | 7613 | 2660 |
| Median-IQR + Huber + $g_t(\boldsymbol{x}_t)$ | **79.6** | **2915** | **1948** | 80.1 | **961** | **700** | **33.3** | **10211** | 3884 | **72.5** | **4111** | **2565** | **24.5** | **7217** | **2327** |

subsequent paragraph. The one-sided $p \approx 0.032$ suggests that, at the 5% significance level, FlowSteer uses fewer tokens than the strongest baseline when achieving comparable accuracy.

**FlowSteer prioritizes efficient reasoning over mere brevity** A deeper analysis of token usage shows a key advantage of our method. On AIME24 and LiveCodeBench, while methods like AlphaOne often produce shorter responses on average (column "T." in Table 2), this brevity does not consistently translate to the most concise reasoning for *correct* answers ("T.@C"). In contrast, FlowSteer requires fewer tokens to generate correct solutions on these benchmarks, as shown in Table 2 and Figure 4. This suggests that FlowSteer does not merely shorten outputs, but more effectively streamlines the underlying reasoning paths required for a successful outcome. This efficiency is particularly noteworthy given that FlowSteer successfully solves harder problems. As our study in Section C shows, problems correctly solved by FlowSteer are up to **1.5× more difficult** than those solved by AlphaOne. These challenging questions naturally demand longer CoTs, which can increase the average token count. Thus, the superior performance of FlowSteer is rooted in adaptively allocating necessary reasoning to complex problems rather than compressing all reasoning paths.

## 4.4 Ablation study

**Median-IQR normalization and Huber loss are necessary** FlowSteer relies on two key components: robust training techniques and probabilistic guidance. The robust training techniques, specifically median-IQR normalization and Huber loss, are prerequisites for stability. Without them, the flow model is poorly fitted, causing the ODE solver to produce divergent steering trajectories during inference. This leads to unusable outputs and the trivial accuracy observed in Table 3.

**Probabilistic guidance substantially boosts performance** As shown in Table 3, even without guidance, the base flow model improves accuracy in all benchmarks except LiveCodeBench and reduces token usage in all benchmarks except AIME24, relative to the vanilla LRM. Adding probabilistic guidance yields a further substantial boost, consistently lowering token counts and increasing accuracy by **up to 20**% absolute improvement (on AIME24). Therefore, by helping representations escape low-velocity zones, the guidance mechanism enables more effective compression of CoTs and unlocks higher accuracy. An additional ablation on the guidance strength $\eta$ is presented in Section D.2.

## 4.5 Analysis on space and time complexity

**Experimental setup** We assess the computational efficiency of FlowSteer in Table 4 across all evaluated datasets. Besides the average accuracy ("Acc.") and token count ("T."), we measure average tokens per second ("TPS"), end-to-end latency per answer ("Lat."), along with relative parameter overhead ("+Prms"). All inference benchmarks were conducted using NVIDIA A100 GPUs with the `HF transformers` library (Wolf et al., 2020).

Table 4: Space and time analysis across LRMs. "Acc.", "T." are the *average* accuracy and token count across all benchmarks, "TPS" is the number of tokens generated per second, "Lat." is the average time per answer in seconds, and "+Prms" is the relative parameter overhead.

| Methods | R1-1.5B | | | | | R1-7B | | | | | QwQ-32B | | | | |
|---|---|---|---|---|---|---|---|---|---|---|---|---|---|---|---|
| | Acc. ↑ | T. ↓ | TPS ↑ | Lat. ↓ | +Prms ↓ | Acc. ↑ | T. ↓ | TPS ↑ | Lat. ↓ | +Prms ↓ | Acc. ↑ | T. ↓ | TPS ↑ | Lat. ↓ | +Prms ↓ |
| Vanilla | 44.2 | 6936 | 39.1 | 177.4 | 0% | 68.5 | 5327 | 37.8 | 140.9 | 0% | 81.4 | 5446 | 27.7 | 196.6 | 0% |
| s1* | 46.4 | 9188 | 39.0 | 235.6 | 0% | 66.2 | 6531 | 37.7 | 173.2 | 0% | 83.8 | 5780 | 27.6 | 209.4 | 0% |
| CoD | 48.1 | 6625 | **39.1** | 169.4 | 0% | 68.3 | 4942 | **37.8** | 130.7 | 0% | 84.2 | 4943 | **27.7** | 178.4 | 0% |
| LogitsPenalty | 46.3 | 6590 | 39.0 | 169.0 | 0% | 67.4 | 5118 | 37.7 | 135.8 | 0% | 82.6 | 5161 | 27.5 | 187.7 | 0% |
| AlphaOne | 52.8 | **4329** | 38.8 | **111.6** | 0% | 71.2 | 4523 | 37.6 | 120.3 | 0% | 80.1 | **4079** | 27.5 | **148.3** | 0% |
| SEAL | 56.5 | 5341 | 38.4 | 139.1 | < 0.1% | 73.1 | 4605 | 37.2 | 123.8 | < 0.1% | 82.4 | 5072 | 27.4 | 185.1 | < 0.1% |
| FlowSteer | **58.0** | 5083 | 38.0 | 133.8 | 3.1% | **74.5** | **4181** | 36.9 | **113.3** | 1.1% | **87.1** | 4922 | 25.9 | 190.0 | 0.6% |

**Analysis of computational overhead and latency**   In Table 4, FlowSteer increases the parameters by only 0.6% to 3.1% compared to the vanilla LRM. The introduction of the Flow Matching module results in a minor reduction in TPS (e.g., a drop of approximately 1.8 TPS on QwQ-32B). However, despite the minor drop in TPS, FlowSteer achieves a reduction in end-to-end latency compared to the vanilla LRM across all model sizes. On R1-7B, FlowSteer achieves the lowest latency among all methods (113.3s) while simultaneously attaining the highest accuracy (74.5%). On R1-1.5B, FlowSteer ranks as the second-fastest method (133.8s), surpassed only by AlphaOne (111.6s), yet it significantly outperforms AlphaOne in accuracy (58.0% vs. 52.8%). On QwQ-32B, while some baseline methods achieve lower latency, they do so at the cost of performance degradation. For example, AlphaOne is faster (148.3s) but drops accuracy to 80.1%, and CoD (178.4s) reaches only 84.2%. In contrast, FlowSteer achieves a dominant 87.1% accuracy while maintaining a latency (190.0s) that is still lower than the Vanilla model (196.6s). Finally, we emphasize that our primary focus in to compress CoTs to achieve an improved accuracy-token count trade-off. Whereas reductions in token count represent a fundamental algorithmic efficiency, latency and TPS are implementation-dependent metrics. Optimizations such as employing advanced ODE solvers (Lu et al., 2025) or faster Flow Matching models (Geng et al., 2025; Lu & Song, 2025; Frans et al., 2025) might further enhance latency and throughput without compromising accuracy. We leave such explorations for future work.

## 5   Related work

**Concise reasoning for LRMs**   Two main directions have been explored to promote conciseness in LRMs. A first line of work focuses on inference-time interventions, which guide model behavior without modifying parameters. These approaches operate at different levels, including prompts (Xu et al., 2025a), tokens (Xia et al., 2025; Muennighoff et al., 2025; Zhang et al., 2025; Wang et al., 2025a), and logits (Wang et al., 2025c; Yang et al., 2025). Our work falls under the subtype of *representation steering* (Park et al., 2025; Li et al., 2023; Liu et al., 2024), which directly manipulates hidden states. While prior steering methods (Chen et al., 2025a; Zhao et al., 2025; Azizi et al., 2025; Huang et al., 2025; Sheng et al., 2025; Eisenstadt et al., 2025) have successfully improved reasoning efficiency, they typically apply a uniform steering vector, effectively modeling a linear translation. FlowSteer extends this by learning a nonlinear velocity field, offering the expressivity needed to capture complex distributional shifts that linear approximations may not fully address. A complementary line of work uses Reinforcement Learning (RL) to fine-tune LRMs for producing shorter CoTs (Kang et al., 2025; Munkhbat et al., 2025; Chen et al., 2025b; Hou et al., 2025; Qiao et al., 2025; Dai et al., 2025; Yuan et al., 2025; Xia et al., 2025; Luo et al., 2025; Xu et al., 2025b; Fang et al., 2025; Xiang et al., 2025). These two lines are complementary and address different trade-offs. RL-based methods modify the LRM weights to incentivize concise reasoning without intervening in the inference computation, though this often requires expensive fine-tuning. In contrast, inference-time methods avoid training costs and preserve the original model weights, but necessitate explicit manipulation of inputs, logits, or hidden states during generation. In principle, inference-time interventions could be applied to RL-optimized models to provide additional control.

**Steering as distribution transport**   A growing body of work interprets representation steering through the lens of distribution transport. Recent approaches, such as LinearAcT (Rodriguez et al., 2025), address this by solving for optimal transport using a *linear* transformation. Another relevant development is Truth-Flow (Wang et al., 2025b), which grounds steering in Flow Matching to enhance model truthfulness. Our work distinguishes itself from these approaches in both methodology and application. Unlike LinearAcT that adopts close-form linear mapping, we leverage Flow Matching to parameterize a nonlinear velocity field. While TruthFlow shares the Flow Matching foundation, it is tailored for truthfulness; in contrast, FlowSteer is designed to compress LRM reasoning paths. The distinct goals and challenges necessitate specific technical innovations, including different training objectives and a specialized inference-time guidance mechanism suited for reasoning efficiency.

**Flow Matching**   Our approach is grounded in Flow Matching (Lipman et al., 2023; Liu, 2022; Lipman et al., 2024), a powerful generative modeling technique successful in domains like image and molecule generation (Labs, 2024; Labs et al., 2025; Esser et al., 2024; Campbell et al., 2024). Additionally, guidance mechanism is often added to such models to align outputs with conditions like text prompts (Dhariwal & Nichol, 2021; Ho & Salimans, 2021; Karras et al., 2024).

## 6   Conclusion

We introduce FlowSteer, a steering framework leveraging Flow Matching to facilitate efficient reasoning paths in LRMs. By learning a nonlinear velocity field for distributional alignment, our method overcomes the limitations of linear steering. We solve key challenges in applying flows to LRM activations with robust training and a specialized guidance mechanism. Empirical results across various reasoning tasks and model scales support the effectiveness of our approach, demonstrating that FlowSteer effectively reduces the discrepancy between the steered and target representation distributions. This alignment translates into improved accuracy and more concise CoTs, particularly on challenging reasoning benchmarks.

While FlowSteer demonstrates strong performance, several avenues remain for further refinement. For instance, our current guidance mechanism utilizes a Gaussian approximation with fixed strength; future iterations could explore non-Gaussian or adaptive alternatives to further enhance steering precision. Additionally, while we focused on the SEAL intervention protocol, extending FlowSteer to other intervention settings represents a meaningful direction. Regarding computational overhead, solving the ODE for steering naturally incurs extra evaluation steps than linear operations. However, we anticipate this gap can be bridged by integrating recent advances in accelerating Flow Matching (Geng et al., 2025; Frans et al., 2025).

Looking forward, while linear steering has been widely adopted and proven effective for many applications, our work suggests that a probabilistic, distributional alignment perspective offers a valuable step forward. By grounding steering in Flow Matching, FlowSteer provides a flexible tool for manipulating LLM representations between distributions associated with contrastive behaviors. This capability opens promising directions beyond efficient reasoning, such as improving model alignment or mitigating hallucination. We hope this work inspires further exploration into nonlinear, theoretically grounded interventions that not only improve efficiency but also ensure finer controllability and greater trustworthiness in large language models.

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

# A   Implementation details

## A.1   Training Data Construction

**Intervention protocol alignment with SEAL**   To ensure a fair and direct comparison, in Section 4 we adopt the intervention protocol from SEAL (Chen et al., 2025a). Furthermore, we use the identical training data for both FlowSteer and SEAL. Specifically, we follow the data extraction procedure outlined by SEAL to create a shared dataset of hidden representations from the MATH/train dataset. This dataset is then used for two distinct purposes: (1) to compute the single, global steering vector $\boldsymbol{v}$ for SEAL, and (2) to train the nonlinear flow model $v_{\boldsymbol{\theta}}$ for FlowSteer.

A critical aspect of the intervention protocol is consistency in the intervention layer. The layer from which representations are extracted during training must be the same layer that is steered during inference. We adopt the exact same target layers as SEAL: layer 20 for the R1-1.5B and R1-7B models, and layer 55 for the QwQ-32B model. By keeping all experimental factors consistent except for the steering mechanism itself, we ensure that any observed performance differences can be directly attributed to the design of our nonlinear steering method rather than to confounding variables.

**Source and target set construction**   The extraction process begins by running inference with the vanilla LRMs on the MATH/train dataset. From these outputs, we select a balanced set of correctly and incorrectly solved problems. We use 500 correct and 500 incorrect samples for R1-1.5B, 1200 of each for R1-7B, and 1800 of each for QwQ-32B.

Following SEAL's methodology, we then populate the source and target representation sets. This involves categorizing the intermediate reasoning steps, which are demarcated by "\n\n" tokens, using a collection of keywords (e.g., "Alternatively", "Wait"). This protocol labels each step as one of three types: *Transition*, *Reflection* (verbose, self-correcting thought), or *Execution* (concise, forward-progressing thought). The hidden state corresponding to the "\n\n" token at the end of each step is selected as that step's representative hidden state. Representations from steps labeled as "Reflection" or "Transition" are added to the source set $\mathcal{S}$ (representing potentially verbose reasoning), while representations from "Execution" steps are added to the target set $\mathcal{T}$ (representing concise reasoning). This process often yields multiple source and target representations from a single problem. The final counts of the source-target representation pairs used for training are provided in Table 5.

## A.2   Flow model architecture

We implement the flow model as a Multi-layer Perceptron (MLP) with ReLU as activation function. The MLP accepts a $(d+1)$-dimensional input, which is formed by concatenating the $d$-dimensional hidden representation from an LRM layer with the scalar time step $t \in [0, 1]$. The network's output is a $d$-dimensional vector, matching the dimension of the input representation. Additional architectural details, such as the number of layers and intermediate dimensions, are provided in Table 5.

## A.3   Training and inference configurations

Detailed hyperparameters for both training and inference are provided in Table 5. For the training phase, this includes optimizer settings, batch size, and the total number of iterations, etc. We highlight the efficiency of our approach: even the flow model for QwQ-32B can be trained on a single NVIDIA A100 GPU in under 24 hours.

For inference-time steering, the table specifies the ODE solver, the target LRM layer, and the solver's tolerances. Specifically, `rtol` and `atol` denote the relative and absolute tolerances for the adaptive ODE solver, respectively.

## A.4   Hyperparameter sweep

To ensure a fair comparison, we perform a hyperparameter sweep for both the probabilistic guidance strength $\eta$ of FlowSteer and the linear strength $\gamma$ of SEAL over the grid $\{0.85, 0.90, 1.00, 1.10, 1.15\}$. For each bench-

Table 5: Implementation details of the flow models, including the configurations of the architecture, training and inference. In "Architecture", the "representation dimensions" refer to the hidden representation dimensions of the hosting LRM. In "ODE solver", the "rtol" denotes the relative tolerance, and the "atol" denotes the absolute tolerance.

|  | Configuration | *R1-1.5B* | *R1-7B* | *QwQ-32B* |
|---|---|---|---|---|
| Architecture | Representation dimensions | 1536 | 3584 | 5120 |
|  | Intermediate dimensions | 3072 | 3584 | 5120 |
|  | Total (linear) layers | 6 | 6 | 8 |
| Training | Question samples | 1000 | 2400 | 3600 |
|  | Source-target representation pairs | 54,796 | 78,980 | 63,705 |
|  | Learning rate | 0.0001 | 0.0001 | 0.0001 |
|  | Optimizer | `Adam(betas=[0.90, 0.95])` | | |
|  | Weight decay | 0 | 0 | 0 |
|  | Clip gradients | False | False | False |
|  | Learning rate decay | False | False | False |
|  | Iterations | $1.7120 \times 10^6$ | $2.9616 \times 10^6$ | $2.7860 \times 10^6$ |
|  | Batch size | 32 | 32 | 32 |
| Inference | ODE solver | `Dopri5(rtol=0.001, atol=0.001)` | | |
|  | Steering LRM layer | 20 | 20 | 55 |

mark, 25% of the evaluation samples are held out as a validation set for the hyperparameter search. We then report the best-performing results for both FlowSteer and SEAL, with the optimal values for $\eta$ and $\gamma$ detailed in Table 6 and Table 7, respectively.

As shown in Table 6, the value $\eta = 1.0$ is a robust choice that performs best in a majority of settings. In Section D.2, we provide the ablation study on the guidance strength $\eta$.

Table 6: The chosen values of the guidance strength $\eta$ in FlowSteer.

|  | MATH500 | GSM8K | AIME24 | AMC23 | LiveCodeBench |
|---|---|---|---|---|---|
| *R1-1.5B* | 0.9 | 1.0 | 0.85 | 1.0 | 1.0 |
| *R1-7B* | 1.0 | 1.0 | 1.0 | 1.15 | 0.9 |
| *QwQ-32B* | 1.0 | 1.15 | 1.0 | 1.0 | 1.0 |

Table 7: The chosen values of the linear strength $\gamma$ for SEAL.

|  | MATH500 | GSM8K | AIME24 | AMC23 | LiveCodeBench |
|---|---|---|---|---|---|
| *R1-1.5B* | 1.0 | 1.0 | 1.0 | 0.9 | 1.15 |
| *R1-7B* | 1.0 | 0.85 | 1.0 | 1.10 | 1.0 |
| *QwQ-32B* | 1.0 | 1.0 | 0.85 | 1.0 | 1.0 |

Table 8: Most accurate baseline used for token-usage comparison with FlowSteer in each model–dataset setting. If the absolute accuracy gap between FlowSteer and the most accurate baseline is at least 5 percentage points, the setting is omitted from the paired analysis and marked as $|\Delta\text{Acc}| \geq 5$.

| Model | MATH500 | GSM8K | AIME24 | AMC23 | LiveCodeBench |
|---|---|---|---|---|---|
| R1-1.5B | SEAL | SEAL | SEAL | $|\Delta\text{Acc}| \geq 5$ | AlphaOne |
| R1-7B | SEAL | AlphaOne | Vanilla | SEAL | SEAL |
| QwQ-32B | SEAL | SEAL | $|\Delta\text{Acc}| \geq 5$ | $|\Delta\text{Acc}| \geq 5$ | SEAL |

## B    Statistical test of token usage

In this section, we conduct a statistical test on the token counts ("T." in Table 2) of FlowSteer against other baselines. A direct comparison of token counts can be misleading, as it ignores the entanglement between accuracy and token usage; tackling difficult questions to attain higher accuracy naturally requires longer CoTs, as discussed in Section 4.3 and Section C. Therefore, we narrow our focus to test the hypothesis that, *under comparable accuracy*, FlowSteer generates *fewer tokens* than the best-performing baseline.

Concretely, for every LRM and benchmark combination in Table 2, we first identify the most accurate baseline method. We then retain only those settings where the absolute accuracy difference between the baseline and FlowSteer is within 5 percentage points; otherwise, the setting is omitted, as it does not represent a fair "equal-accuracy" comparison. This procedure yields 12 out of 15 configurations, with the selected strongest baselines outlined in Table 8. For each retained comparison pair (strongest baseline vs. FlowSteer), we compute the average number of generated tokens across all answers, forming per-setting differences $d_i = T_{\text{baseline},i} - T_{\text{FlowSteer},i}$. We then apply a one-sided Wilcoxon signed-rank test with the null hypothesis that the median difference is zero and the alternative hypothesis that FlowSteer uses fewer tokens ($\text{median}(d_i) > 0$). Across the 12 settings, FlowSteer uses fewer tokens in 10 cases, and the Wilcoxon test yields a one-sided $p \approx 0.032$. This result is statistically significant at the 5% level and supports our claim that, when achieving comparable accuracy, FlowSteer consistently improves efficiency by reducing token usage relative to strong baselines.

## C    A nuanced analysis of question difficulty and token usage on AIME24

In this section, we provide a more nuanced analysis of the results in Table 2 to explain why FlowSteer consumes more tokens overall than AlphaOne on AIME24 despite achieving higher accuracy. We hypothesize that this is because FlowSteer excels at solving more difficult problems, which inherently require longer solutions.

To investigate this, we first define a **difficulty score** for each question. Our metric uses the token length in the **ground truth (GT)** solution as a proxy for difficulty. The longer the GT solution, the harder the problem. We normalize this value to derive the score $Q^{(i)}$ for the $i$-th question as follows:

$$Q^{(i)} = \frac{T_{\text{GT}}^{(i)}}{T_{\text{GT}}^{\min}}, \tag{9}$$

where $T_{\text{GT}}^{(i)}$ is the token length of its GT solution, and $T_{\text{GT}}^{\min}$ is the minimum GT token length across all questions in the AIME24 dataset. We then calculate the average difficulty score of all questions a model answers correctly.

As shown in Table 9, FlowSteer consistently solves questions with the highest average difficulty score (column "Difficulty@C"). The average difficulty score of FlowSteer is $1.1 \sim 1.5\times$ higher than that of AlphaOne. Since our difficulty metric is proportional to the GT solution length, these more challenging problems naturally demand more tokens to be solved correctly. This evidence suggests that FlowSteer's higher token

Table 9: Performance on the AIME24 benchmark, analyzing accuracy, token efficiency, and the **difficulty of solved problems**. We report Pass@1 accuracy (**Acc.** ↑), average generated tokens across all answers (**T.** ↓), average tokens in correct answers (**T.@C** ↓), and the average difficulty of correctly answered questions (**Difficulty@C** ↑). The results show that AlphaOne uses the fewest tokens overall on the AIME24 benchmark. However, FlowSteer consistently achieves superior accuracy by successfully solving more complex problems, as evidenced by its higher average difficulty score in solved problems.

| Methods | Acc. ↑ | T. ↓ | T.@C ↓ | Difficulty@C ↑ |
|---|---|---|---|---|
| *DeepSeek-R1-Distill-Qwen-1.5B* | | | | |
| Vanilla | 6.7 | 13807 | **2735** | 1.47 |
| AlphaOne | 23.3 | 7314 | 6304 | 2.16 |
| **FlowSteer** | **33.3** | 10211 | 3884 | **2.48** |
| *DeepSeek-R1-Distill-Qwen-7B* | | | | |
| Vanilla | 50.0 | 9903 | 5539 | 2.61 |
| AlphaOne | 46.7 | **7605** | 6052 | 2.28 |
| **FlowSteer** | **53.3** | 8453 | **4132** | **2.75** |
| *Qwen-QwQ-32B* | | | | |
| Vanilla | 63.3 | 10791 | 8486 | 3.10 |
| AlphaOne | 53.3 | **6544** | **5165** | 2.56 |
| **FlowSteer** | **76.7** | 9848 | 8355 | **3.93** |

consumption is a direct result of its superior ability to tackle complex problems that require more elaborate reasoning.

# D   Additional study

## D.1   Probabilistic guidance with various model scales

In this subsection, we present a comprehensive ablation study to demonstrate the effectiveness of probabilistic guidance, verifying that its benefits generalize to larger models. The results are summarized in Table 10.

Having already analyzed the guidance mechanism on the R1-1.5B model in Section 4.4, here we focus on larger model scales. First, on the R1-7B model, adding probabilistic guidance (compared to the baseline in row "w/o $g_t(\boldsymbol{x}_t)$") improves both accuracy (column "Acc.") and average token usage (column "T.") across all five benchmarks. This is highlighted by a significant **12.5% absolute accuracy improvement** on AMC23. Next, We test if this advantage further generalizes to QwQ-32B model. The performance gains persist, with accuracy increasing on all five benchmarks and token usage improving on four. Notably, the guidance still achieves a substantial **10.1% accuracy improvement** on AIME24 compared to the setting without guidance. These results confirm that the enhancements from our probabilistic guidance do not diminish as model scale increases from 1.5B to 32B, highlighting the method's broad effectiveness and scalability to different model sizes.

## D.2   Ablation study on guidance strength

In this subsection, we quantitatively analyze the effect of the probabilistic guidance strength, $\eta$. We evaluate the accuracy and average token count on the MATH500 and LiveCodeBench benchmarks across an expanded range of $\eta \in \{0.1, 0.85, 0.9, 1.0, 1.1, 1.15, 1.5, 2.0\}$. The results are visualized in Figure 5, which presents

Table 10: Ablation of the probabilistic guidance on the *R1-1.5B*, *R1-7B*, and *QwQ-32B* LRMs.

| Methods | MATH500 | | | GSM8K | | | AIME24 | | | AMC23 | | | LiveCodeBench | | |
|---|---|---|---|---|---|---|---|---|---|---|---|---|---|---|---|
| | Acc. | T. | T.@C | Acc. | T. | T.@C | Acc. | T. | T.@C | Acc. | T. | T.@C | Acc. | T. | T.@C |
| *DeepSeek-R1-Distill-Qwen-1.5B* | | | | | | | | | | | | | | | |
| Vanilla | 66.6 | 4785 | 2390 | 73.8 | 2072 | 819 | 6.7 | 13807 | **2735** | 52.5 | 6269 | 2998 | 21.2 | 7749 | 2372 |
| w/o $g_t(\boldsymbol{x}_t)$ | 78.6 | 3168 | 2026 | **81.2** | 1071 | 782 | 13.3 | 11534 | 2996 | 67.5 | 4688 | 2919 | 21.0 | 7613 | 2660 |
| w/ $g_t(\boldsymbol{x}_t)$ | **79.6** | **2915** | **1948** | 80.1 | **961** | **700** | **33.3** | **10211** | 3884 | **72.5** | **4111** | **2565** | **24.5** | **7217** | **2327** |
| *DeepSeek-R1-Distill-Qwen-7B* | | | | | | | | | | | | | | | |
| Vanilla | 87.0 | 3341 | 2605 | 87.9 | 1160 | 867 | 50.0 | 9903 | 5539 | 72.5 | 5431 | 3698 | 45.3 | 6799 | 3394 |
| w/o $g_t(\boldsymbol{x}_t)$ | 90.0 | 2663 | 2236 | 88.3 | 803 | 734 | 53.3 | 8543 | **3737** | 77.5 | 4068 | **2627** | 48.3 | 6145 | **3128** |
| w/ $g_t(\boldsymbol{x}_t)$ | **90.2** | **2549** | **2010** | **88.6** | **797** | **732** | **53.3** | **8453** | 4132 | **90.0** | **3177** | 2739 | **50.5** | **5930** | 3204 |
| *Qwen QwQ-32B* | | | | | | | | | | | | | | | |
| Vanilla | 90.8 | 3549 | 3027 | 95.6 | 1157 | 1069 | 63.3 | 10791 | 8486 | 80.0 | 6053 | 5067 | 77.5 | 5680 | 4494 |
| w/o $g_t(\boldsymbol{x}_t)$ | 90.2 | 3189 | **2655** | 96.1 | 968 | 886 | 66.6 | 9932 | **7398** | 85.0 | **5463** | 4662 | 79.5 | 5081 | **3879** |
| w/ $g_t(\boldsymbol{x}_t)$ | **91.0** | 3118 | 2675 | **96.5** | **917** | **857** | **76.7** | 9848 | 8355 | **90.0** | 5698 | 5220 | **81.2** | **5030** | 3970 |

accuracy as line plots and average token count as bar charts. For reference, the performance of the vanilla LRM baseline is included.

Our analysis yields the following observations:

- **Broad effective range:** Our method demonstrates robust performance across a wide spectrum of guidance strengths. Substantial improvements in both accuracy and token reduction persist even at the boundaries of $\eta = 0.1$ and $\eta = 1.5$.

- **Low guidance regime** ($\eta = 0.1$)**:** At low guidance strengths, the method remains stable but operates with reduced efficiency. While accuracy is generally maintained near peak levels, the token reduction is smaller compared to the optimal range. We hypothesize that this is because the magnitude of the probabilistic guidance is insufficient to effectively push representations out of the "low-velocity zones" described in Section 3, resulting in partial stagnation and longer reasoning paths.

- **Optimal range** ($\eta \in [0.85, 1.15]$)**:** This range consistently yields the most favorable trade-off, achieving peak accuracy while simultaneously minimizing token usage.

- **High guidance regime** ($\eta = 2.0$)**:** At the highest strength tested, we observe a degradation in performance. While token counts are still reduced, accuracy often declines towards the vanilla baseline. This aligns with our derivation in Eq. (8): at very high $\eta$, the guidance term dominates the ODE dynamics, suppressing the flow model's contribution. This "over-steering" forces the trajectory to ignore the fine-grained structure of the target representation distribution, leading to suboptimal reasoning performance.

In summary, our probabilistic guidance demonstrates a clear advantage over the baseline that is robust in the choice of $\eta$. The behavior at the extremes ($\eta = 0.1$ and $\eta = 2.0$) is consistent with our theoretical expectations regarding the interplay between the learned flow field and the guidance term.

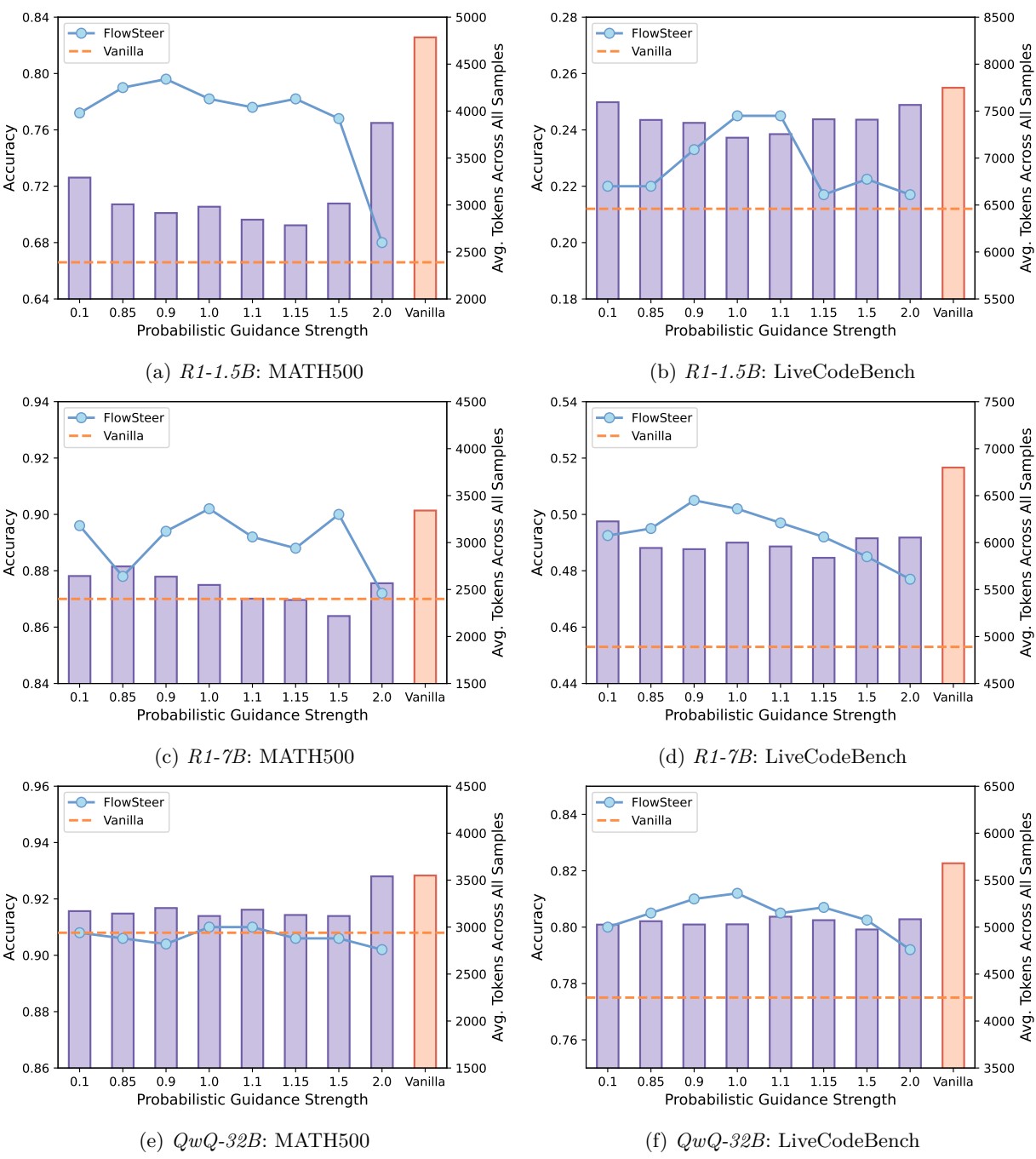

(a) *R1-1.5B*: MATH500

(b) *R1-1.5B*: LiveCodeBench

(c) *R1-7B*: MATH500

(d) *R1-7B*: LiveCodeBench

(e) *QwQ-32B*: MATH500

(f) *QwQ-32B*: LiveCodeBench

Figure 5: The line plots report accuracy, while the bar plots show the average token count across all samples. The dashed line and the rightmost bar correspond to the vanilla LRM baseline. For visual clarity, the bars are evenly spaced along the $x$-axis, although the underlying hyperparameter grid is uneven.

# E   Distributional distance metrics

## E.1   A brief introduction to MMD, FID, and KID

To quantitatively evaluate the alignment between the steered representations and the target representations, we employ three established metrics in Section 4.2. Each provides a different lens through which to measure the distance between two sets of samples.

**Maximum Mean Discrepancy (MMD)** (Gretton et al., 2012) is a non-parametric measure that compares two distributions by mapping their samples into a high-dimensional Reproducing Kernel Hilbert Space (RKHS). If the distributions are identical, their mean embeddings in this space coincide. MMD computes the distance between these mean embeddings, with larger values indicating greater discrepancy. Its strength lies in its generality and strong theoretical guarantees, as it does not impose assumptions on the underlying data distributions.

**Fréchet Inception Distance (FID)** (Heusel et al., 2017), originally proposed for evaluating generative image models, compares distributions of high-level features extracted by a pre-trained network. Assuming these feature distributions follow multivariate Gaussians, FID computes the **Fréchet distance** (i.e., the Wasserstein-2 distance) between them, incorporating both mean and covariance. .

**Kernel Inception Distance (KID)** (Sutherland et al., 2018) combines ideas from MMD and FID to address limitations of the latter. Like FID, it relies on features from a neural network, but instead of assuming Gaussianity, KID applies the MMD framework with a polynomial kernel. This yields an unbiased estimator that is often more stable than FID, particularly with limited sample sizes. As with FID, lower KID scores indicate smaller discrepancies between the distributions.

## E.2   Implementation details of the distance metrics

Since FID and KID were originally designed for image generation, they typically rely on a pre-trained network to extract high-level features. In our scenario, however, we apply these metrics directly to the hidden representations from LRMs. These representations already contain rich semantic information, making an external feature extractor unnecessary.

Below are the specific implementation details for each metric:

- **MMD:** We use the implementation from `pytorch-ignite` (Fomin et al., 2020). The performance of MMD is sensitive to its kernel variance hyperparameter, `var`. Setting it too large or small can make the metric insensitive. We determine this value empirically using a heuristic: we set `var` to be on the same order of magnitude as the squared norm of SEAL's steering vector, $||\boldsymbol{v}||_2^2$, as this value represents the squared distance between the empirical distribution means. The specific values used are 4,000 (for settings with R1-1.5B), 7,000 (R1-7B), and 26,000 (QwQ-32B). For readability in Table 1, all reported MMD scores are scaled by a factor of 1,000.

- **FID:** We also use the implementation provided by `pytorch-ignite` (Fomin et al., 2020).

- **KID:** Our implementation is adapted from `torchmetrics` (Detlefsen et al., 2022). We employ a polynomial kernel and use its default parameters: a degree (`degree`) of 3 and a coefficient (`coef`) of 1.0.

# F   Examples generated by the steered models

The example below contrasts the output of the vanilla model with that of the model steered by FlowSteer. The vanilla model exhibits *over-reflection*, causing it to get stuck in a repetitive reasoning loop. This behavior ultimately leads to the model exhausting the context and hitting the maximum token limit of 10,000 before finding a solution. In contrast, the steered model produces a clear and efficient reasoning path that arrives at the final correct answer, using only 1,367 tokens.

---

**Example 1 with FlowSteer R1-1.5B; Tokens spent: 1,367**

**Question Prompt:**
Answer the following questions. You should think step-by-step and put your final answer within
\boxed{}. Question: For some real numbers $a$ and $b$, the equation

$$8x^3 + 4ax^2 + 2bx + a = 0$$

has three distinct positive roots. If the sum of the base-2 logarithms of the roots is 5, what is the
value of $a$?

---

**LRM's Answer:**
<think> Okay, so I have this problem here: there's a cubic equation $8x^3 + 4ax^2 + 2bx + a = 0$, and
it's given that it has three distinct positive roots. Also, the sum of the base-2 logarithms of the
roots is 5. I need to find the value of $a$. Hmm, let me think about how to approach this.
First, I remember that for a cubic equation of the form $px^3 + qx^2 + rx + s = 0$, the sum of the roots,
the sum of the product of the roots two at a time, and the product of the roots can be found using
Vieta's formulas. Maybe I can apply that here.
Let me denote the roots as $r$, $s$, and $t$. Since all roots are positive, that might help later on.
According to Vieta's formulas for a cubic equation $px^3 + qx^2 + rx + s = 0$, we have:
1. $r + s + t = -\frac{q}{p}$ 2. $rs + rt + st = \frac{r}{p}$ 3. $rst = -\frac{s}{p}$
In our case, the equation is $8x^3 + 4ax^2 + 2bx + a = 0$. So, comparing to the general form, $p = 8$,
$q = 4a$, $r = 2b$, and $s = a$.
Applying Vieta's formulas:
1. Sum of roots: $r + s + t = -\frac{4a}{8} = -\frac{a}{2}$ 2. Sum of products two at a time: $rs + rt + st = \frac{2b}{8} = \frac{b}{4}$ 3.
Product of roots: $rst = -\frac{a}{8}$
We are also given that the sum of the base-2 logarithms of the roots is 5. That is:

$$\log_2 r + \log_2 s + \log_2 t = 5$$

Using logarithm properties, this can be rewritten as:

$$\log_2(r \cdot s \cdot t) = 5$$

Which implies:

$$r \cdot s \cdot t = 2^5 = 32$$

From Vieta's formula, we already have $rst = -\frac{a}{8}$. So:

$$-\frac{a}{8} = 32$$

Solving for $a$:

$$-\frac{a}{8} = 32 \implies a = -32 \times 8 = -256$$

So, is $a = -256$? Wait, let me double-check that.
Product of roots is $rst = 32$, and from Vieta, $rst = -\frac{a}{8}$. So:

$$-\frac{a}{8} = 32 \implies a = -256$$

Yes, that seems correct. But let me make sure that this is consistent with the other Vieta's formulas.
We have $r + s + t = -\frac{a}{2}$. If $a = -256$, then:

$$r + s + t = -\frac{-256}{2} = 128$$

So, the sum of the roots is 128. That's a pretty large number, but since all roots are positive, that's possible.

Next, the sum of the products two at a time is $rs + rt + st = \frac{b}{4}$. But we don't know $b$, so maybe we don't need it. However, let's see if we can find $b$ or if it's not necessary for the answer.

We were given that all roots are distinct and positive. So, the cubic must have three distinct positive roots, which is another condition. But since we only need to find $a$, and we've found $a = -256$, maybe that's sufficient.

Let me recap:

- The product of the roots is 32, which is positive, so that's consistent with all roots being positive.
- The sum of the roots is 128, which is positive, so that's also consistent. - The sum of the products two at a time is $b/4$, but since we don't have any information about $b$, we don't need to find it.

So, unless there's a mistake in my reasoning, $a = -256$ should be the correct answer.

Given that, I think I'm confident with this result.

**Final Answer** The value of $a$ is $\boxed{-256}$. </think>

Given the cubic equation $8x^3 + 4ax^2 + 2bx + a = 0$ with three distinct positive roots, we need to find the value of $a$.

Using Vieta's formulas, we know: 1. The sum of the roots $r + s + t = -\frac{4a}{8} = -\frac{a}{2}$. 2. The product of the roots $rst = -\frac{a}{8}$.

We are also given that the sum of the base-2 logarithms of the roots is 5:

$$\log_2 r + \log_2 s + \log_2 t = 5$$

This can be rewritten using logarithm properties as:

$$\log_2(r \cdot s \cdot t) = 5$$

Thus, the product of the roots is:

$$r \cdot s \cdot t = 2^5 = 32$$

From Vieta's formula, we have:

$$rst = -\frac{a}{8}$$

Equating the two expressions for the product of the roots:

$$-\frac{a}{8} = 32$$

Solving for $a$:

$$a = -256$$

Thus, the value of $a$ is $\boxed{-256}$.

---

**Ground Truth Answer:**

-256

**Example 1 with vanilla R1-1.5B; Tokens spent: 10,000**

**Question Prompt:**
Answer the following questions. You should think step-by-step and put your final answer within
\boxed{}. Question: For some real numbers $a$ and $b$, the equation

$$8x^3 + 4ax^2 + 2bx + a = 0$$

has three distinct positive roots. If the sum of the base-2 logarithms of the roots is 5, what is the
value of $a$?

---

**LRM's Answer:**
<think> Okay, so I have this problem here: there's a cubic equation $8x^3 + 4ax^2 + 2bx + a = 0$, and
it's given that it has three distinct positive roots. Also, the sum of the base-2 logarithms of the
roots is 5. I need to find the value of $a$. Hmm, let me think about how to approach this.
First, I remember that for a cubic equation of the form $px^3 + qx^2 + rx + s = 0$, the sum of the roots,
the sum of the product of the roots two at a time, and the product of the roots can be found using
Vieta's formulas. Maybe I can apply that here.
Given the equation $8x^3 + 4ax^2 + 2bx + a = 0$, let me denote the roots as $r$, $s$, and $t$. Since all roots
are positive, that might help later on.
According to Vieta's formulas for a cubic equation $px^3 + qx^2 + rx + s = 0$, we have:
1. $r + s + t = -\frac{q}{p}$ 2. $rs + rt + st = \frac{r}{p}$ 3. $rst = -\frac{s}{p}$
In our case, $p = 8$, $q = 4a$, $r = 2b$, and $s = a$. So plugging these into Vieta's formulas:
1. $r + s + t = -\frac{4a}{8} = -\frac{a}{2}$ 2. $rs + rt + st = \frac{2b}{8} = \frac{b}{4}$ 3. $rst = -\frac{a}{8}$
Wait, hold on. The product of the roots $rst$ is equal to $-\frac{s}{p}$, which in this case is $-\frac{a}{8}$. But since all
roots are positive, their product should be positive. However, $-\frac{a}{8}$ is positive only if $a$ is negative.
So that tells me that $a$ must be negative. That's an important point.
Also, the sum of the roots $r + s + t = -\frac{a}{2}$. Since $a$ is negative, this sum becomes positive. So, the
sum of the roots is positive, which makes sense because all roots are positive.
The sum of the base-2 logarithms of the roots is given as 5. That is, $\log_2 r + \log_2 s + \log_2 t = 5$. I
remember that the sum of logarithms is the logarithm of the product, so this can be rewritten as
$\log_2(rst) = 5$. Therefore, $rst = 2^5 = 32$.
But from Vieta's formula, we also have $rst = -\frac{a}{8}$. So, setting these equal:
$-\frac{a}{8} = 32$
Solving for $a$:
Multiply both sides by 8: $-a = 256$
Then, multiply both sides by -1: $a = -256$
Wait, that seems straightforward. Let me double-check to make sure I didn't make any mistakes.
So, if $a = -256$, then the product of the roots $rst = -\frac{a}{8} = -\frac{-256}{8} = 32$, which matches the given
condition. So that seems correct.
But just to be thorough, let me check the other Vieta's formulas to see if they hold with $a = -256$.
First, the sum of the roots $r + s + t = -\frac{a}{2} = -\frac{-256}{2} = 128$. So, the sum of the roots is 128.
The sum of the product of the roots two at a time is $\frac{b}{4}$. Hmm, but we don't know $b$, so maybe we
don't need to find it. The problem doesn't give us any information about $b$, so perhaps we don't
need to worry about that.
But just to make sure, let's see if the cubic equation with $a = -256$ and some $b$ would have three
distinct positive roots. Since the problem states that it does, so as long as the cubic equation is
constructed correctly, it should have three distinct positive roots.
Wait, but do we need to ensure that the cubic equation actually has three distinct positive roots?
Because sometimes, even if the product is positive, the roots might not be distinct or might not be
positive. But the problem says it does have three distinct positive roots, so perhaps we don't need
to worry about that.

But just to be safe, let me think about the discriminant of the cubic equation. The discriminant of a cubic equation $ax^3 + bx^2 + cx + d = 0$ is given by:

$\Delta = 18abcd - 4b^3d + b^2c^2 - 4ac^3 - 27a^2d^2$

But in our case, the equation is $8x^3 + 4ax^2 + 2bx + a = 0$. So, let me write the discriminant for this equation.

Let me denote the equation as $8x^3 + 4ax^2 + 2bx + a = 0$. So, coefficients are:

- $p = 8$ - $q = 4a$ - $r = 2b$ - $s = a$

So, the discriminant $\Delta$ is:

$\Delta = 18 \cdot 8 \cdot 4a \cdot 2b \cdot a - 4 \cdot (4a)^3 \cdot a + (4a)^2 \cdot (2b)^2 - 4 \cdot 8 \cdot (2b)^3 - 27 \cdot 8^2 \cdot a^2$

Wait, that seems complicated. Maybe I can compute each term step by step.

First term: $18 \cdot 8 \cdot 4a \cdot 2b \cdot a$

Compute constants: 18 * 8 = 144; 144 * 4 = 576; 576 * 2 = 1152; 1152 * 1 = 1152

Variables: $a * a = a^2$

So, first term is $1152a^2$

Second term: $-4 \cdot (4a)^3 \cdot a$

Compute constants: $4^3 = 64; 64 * 4 = 256; 256 * (-4) = -1024$

Variables: $a^3 * a = a^4$

So, second term is $-1024a^4$

Third term: $(4a)^2 \cdot (2b)^2$

Compute constants: 16 * 4 = 64

Variables: $a^2 * b^2 = a^2b^2$

So, third term is $64a^2b^2$

Fourth term: $-4 \cdot 8 \cdot (2b)^3$

Compute constants: 4 * 8 = 32; 32 * 8 = 256

Variables: $b^3$

So, fourth term is $-256b^3$

Fifth term: $-27 \cdot 8^2 \cdot a^2$

Compute constants: 27 * 64 = 1728

Variables: $a^2$

So, fifth term is $-1728a^2$

Putting it all together:

$\Delta = 1152a^2 - 1024a^4 + 64a^2b^2 - 256b^3 - 1728a^2$

Simplify the constants:

$1152a^2 - 1728a^2 = -576a^2$

So, $\Delta = -576a^2 - 1024a^4 + 64a^2b^2 - 256b^3$

Hmm, that's still complicated. Maybe I can factor out some terms:

Factor out -64:

$\Delta = -64(9a^2 + 16a^4 - a^2b^2 + 4b^3)$

But I don't know if that helps. Maybe I can plug in $a = -256$ and see if the discriminant is positive, which would mean three distinct real roots.

Let me compute each term:

First, $a = -256$, so $a^2 = (-256)^2 = 65536$

Compute each term:

1. $1152a^2 = 1152 * 65536$. Let me compute that:

1152 * 65536: 1152 is 1024 + 128, so 1024 * 65536 = 67108864, and 128 * 65536 = 8388608. So total is 67108864 + 8388608 = 75497472.

2. $-1024a^4 = -1024 * (65536)^2$. Wait, that's a huge number. Maybe I don't need to compute it exactly because it's negative, but I can note that it's a large negative number.

3. $64a^2b^2 = 64 * 65536 * b^2 = 4194304b^2$

4. $-256b^3$

5. $-1728a^2 = -1728 * 65536 = -113462400$

So, putting it all together:

$\Delta = 75497472 - 1024a^4 + 4194304b^2 - 256b^3 - 113462400$

Simplify constants:

75497472 - 113462400 = -37964928

So, $\Delta = -37964928 - 1024a^4 + 4194304b^2 - 256b^3$

Hmm, this is getting too complicated. Maybe I should think differently.

Wait, maybe I don't need to compute the discriminant because the problem only gives information about the sum of logarithms, which directly gives the product of the roots. So, perhaps I don't need to worry about the discriminant because the problem doesn't specify anything else about the roots, like their distinctness or whether they are real or complex. It just says three distinct positive roots, so maybe the discriminant is positive, but I don't need to compute it.

Alternatively, maybe I can use the fact that the equation has three distinct positive roots, so the cubic must cross the x-axis three times, which would require that the function has a local maximum and minimum, and the function values at those points are on opposite sides of the x-axis.

But that might be more involved. Alternatively, maybe I can use the fact that the equation can be scaled or transformed to make it easier.

Wait, another thought: since the product of the roots is 32, and the sum of the roots is 128, maybe I can think of the roots as 32 divided by something, but I don't know.

Alternatively, maybe I can use substitution to make the equation simpler. Let me try substituting $y = 2x$, so that $x = y/2$. Let's see if that helps.

Substituting $x = y/2$ into the equation:

$8(y/2)^3 + 4a(y/2)^2 + 2b(y/2) + a = 0$

Compute each term:

$8 * (y^3/8) = y^3$

$4a * (y^2/4) = ay^2$

$2b * (y/2) = by$

So, the equation becomes:

$y^3 + ay^2 + by + a = 0$

Hmm, that seems a bit simpler. So, the equation is now $y^3 + ay^2 + by + a = 0$. Maybe this is easier to handle.

Now, since the original equation had three distinct positive roots, let's see if the transformed equation also has three distinct positive roots. Since $x = y/2$, if $x$ is positive, then $y$ is positive as well. So, the transformed equation has three distinct positive roots as well.

But I don't know if that helps me find $a$. Maybe I can use Vieta's formulas on the transformed equation.

For the transformed equation $y^3 + ay^2 + by + a = 0$, the sum of the roots $y_1 + y_2 + y_3 = -a$, the sum of the product of roots two at a time $y_1y_2 + y_1y_3 + y_2y_3 = b$, and the product of the roots $y_1y_2y_3 = -a$.

Wait, but in the original equation, the product of the roots was 32, so in the transformed equation, the product is also 32. So, $y_1y_2y_3 = -a = 32$. So, $-a = 32 \implies a = -32$. Wait, that's different from what I got earlier.

Wait, hold on. In the transformed equation, the product of the roots is $-a$, which is equal to 32. So, $-a = 32 \implies a = -32$. But earlier, using the logarithm condition, I got $a = -256$. That's a contradiction. So, which one is correct?

Wait, perhaps I made a mistake in the substitution. Let me double-check.

Original equation: $8x^3 + 4ax^2 + 2bx + a = 0$

Substitute $x = y/2$:

$8 * (y/2)^3 + 4a * (y/2)^2 + 2b * (y/2) + a = 0$

Compute each term:

$8 * (y^3/8) = y^3$

$4a * (y^2/4) = ay^2$

$2b * (y/2) = by$

So, equation becomes:

$y^3 + ay^2 + by + a = 0$

Yes, that's correct. So, the product of the roots in the transformed equation is $-a$, which equals 32, so $a = -32$.

But earlier, using the logarithm condition, I got $a = -256$. So, which is correct?

Wait, perhaps I made a mistake in the substitution. Let me think again.

Wait, in the original equation, the product of the roots is 32, so $rst = 32$. In the transformed equation, the product of the roots is also 32, so $y_1 y_2 y_3 = 32$. But in the transformed equation, the product is $-a$, so $-a = 32 \implies a = -32$.

But in the original equation, the sum of the roots is $r + s + t = -a/2$. So, if $a = -32$, then $r + s + t = -(-32)/2 = 16$. But earlier, from the logarithm condition, I had $r + s + t = 128$. So, that's a contradiction.

Wait, so that suggests that $a = -32$ is not consistent with the sum of the roots being 128. So, perhaps my substitution is incorrect.

Wait, no, hold on. Let me think again. The original equation is $8x^3 + 4ax^2 + 2bx + a = 0$. The transformed equation is $y^3 + ay^2 + by + a = 0$, where $y = 2x$. So, the roots of the transformed equation are $y_1 = 2r$, $y_2 = 2s$, $y_3 = 2t$. So, the sum of the roots in the transformed equation is $y_1 + y_2 + y_3 = 2(r + s + t)$. From Vieta's formula, the sum of the roots in the transformed equation is $-a$. So, $2(r + s + t) = -a$. Therefore, $r + s + t = -a/2$.

But in the original equation, the sum of the roots is $r + s + t = -a/2$. So, if $a = -32$, then $r + s + t = -(-32)/2 = 16$. But earlier, from the logarithm condition, we had $r + s + t = 128$. So, that's a problem.

Wait, so if $a = -32$, the sum of the roots is 16, but the problem says the sum is 128. So, that's inconsistent. Therefore, my substitution must be wrong.

Wait, no, hold on. Let me think again.

*The remaining outputs have been removed to save space. The vanilla model gets stuck in repetitive reasoning loops and reaches the maximum token length limit before finishing its thinking process.*

**Ground Truth Answer:**

-256

