# OpenReview forum: "Steering Large Reasoning Models towards Concise Reasoning via Flow Matching"
_TMLR — Accepted by TMLR_

### Review · Reviewer_iiHA · 2025-12-11

**Summary Of Contributions:**

This paper introduces FlowSteer a new method for reducing the length of reasoning paths in LLMs. The basic idea is to learn how to move internal representation states of the LLM from states associated with longer reasoning paths (”source states”) to states associated with shorter reasoning paths (”target states”). At inference time, states that are likely to produce long reasoning paths can then be transformed using this learned transformation to reduce the reasoning path length.

 Prior work had gone this by identifying a linear transformation between the means of the source and target state distributions. This paper proposes instead to use flow matching to learn a flow that transforms the source distribution into the target distribution. The paper then introduces two tricks needed to make this work in practice: First they find that internal states in LLMs can have large outlier values and propose to address this by using a median-IQR normalization strategy for these states and to use the more robust Huber loss when training the flow model. Second, they find that regions of low velocity exist in the learned velocity field, which they address by adding approximate guidance that pushes samples towards the target distribution in these low velocity areas.

The experiments compare FlowSteer to various baselines on a set of mathematics and coding tasks. The implementation used here uses the same mechanism as SEAL (one of the baselines) to identify on which tokens to intervene, but uses FlowSteer to make the actual change to the internal states at these tokens. Compared to baselines, FlowSteer achieves equal or better answer accuracy while generally using fewer tokens to reach its answers. Ablation studies show that the median-IQR normalization and the use of the Huber loss in training the flow model are both essential to the performance of FlowSteer, and that the guidance is a smaller but still important factor.

**Audience:**

Yes

**Audience Explanation:**

This paper is relevant to people working on LLMs, especially those focused on improving the reasoning efficiency of these models.

**Broader Impact Concerns:**

No concerns beyond those inherent to any work in this area.

**Claims And Evidence:**

Yes

**Claims Explanation:**

The method is described in sufficient detail in the main paper, and the appendix contains the necessary implementation details that would be needed to reproduce the results. FlowSteer is compared to a wide range of recent baselines, but I am not sufficiently familiar with the literature to judge whether any important baselines are missing. Performance is compared on a number of different tasks. The paper also contains ablation studies measuring the contribution of the different parts of FlowSteer to its performance and measurements of computational cost.

**Requested Changes:**

I generally feel this is a strong paper, but the discussion on FlowSteer’s efficiency in correct reasoning (page 9) left me slightly confused, and I would appreciate if the authors could clarify the point they try to make. You point out here that compared to AlphaOne, FlowSteer uses fewer tokens in generating its correct answers, despite using more tokens across all answers. First, I don’t quite understand the relevance of this comparison, considering that the cost of generating the tokens is incurred irrespective of whether it is correct or not. Furthermore, although these results do indeed imply that FlowSteer streamlines reasoning paths towards successful outcomes and thereby uses fewer tokens for these, considering that FlowSteer still uses more tokens overall, does this not also imply at the same time that it uses more tokens to arrive at wrong answers?

---

> ### Author Response · Authors · 2025-12-29
> **Thank you for your feedback**
>
> Dear Reviewer iiHA,
>
> Thank you for your thoughtful and constructive comments on our manuscript.
>
> We have submitted our responses to each point you raised. Please let us know if anything remains unclear or if there are additional issues you would like us to address. We would be happy to elaborate. 🤗
>
> Best regards,
>
> The Authors

---

> > ### Author Response · Authors · 2025-12-29
> > **We have re-submitted the responses to solve potential display problem of OpenReview**
> >
> > Dear Reviewer iiHA,
> >
> > We wanted to let you know that another reviewer encountered a display issue where our responses were not visible in the system. To ensure this does not affect your review, we have re-submitted our responses to your comments as well.
> >
> > We hope this resolves any potential visibility problems. If you also encounter this issue and the problem remains, please feel free to let us know.

---

> ### Author Response · Authors · 2025-12-29
> **Response to Reviewer iiHA**
>
> We thank the reviewer for acknowledging the strength of our work and providing the constructive comments. We address the questions as follows:
>
>
> > Q1.1: Relevance of the additionally comparison of tokens in correct answers (T.@C)?
>
>
> We agree that **T.** (average tokens across all answers) is the primary metric for deployment cost. However, we report **T.@C** not as a deployment-cost proxy, but as a **diagnostic tool** to analyze how different methods achieve low T.  Low T. can stem from "cheap failures" (short, incorrect answers). T.@C isolates successful trajectories to verify if a method actually streamlines reasoning. For instance, on R1-1.5B/AIME24, FlowSteer achieves higher accuracy (33.3% vs. 23.3%) with significantly lower T.@C (3884 vs. 6304) than AlphaOne. This confirms FlowSteer removes redundant steps in valid solutions rather than just cutting output length.
>
> In response, we have updated the title of the second paragraph (highlighted in blue) in Section 4.3 to make the key takeaway more specific.
>
>
> > Q1.2: To reach higher accuracy, FlowSteer uses more tokens overall compared to methods like AlphaOne, and uses less tokens in correct answers, does it imply that it uses more tokens to arrive at wrong answers?
>
> 1. The reviewer is correct that FlowSteer generates more tokens for incorrect answers compared to AlphaOne.
> 2. We view this as a necessary trade-off for **adaptive reasoning allocation**. AlphaOne enforces brevity on all traces by penalizing the generation of 'Wait' tokens, often causing the model to collapse prematurely on complex problems ('fast failures'). In contrast, FlowSteer and methods like SEAL lack this hard constraint by design, allowing them to explore longer reasoning paths—even if this results in extended traces for incorrect answers. Preserving this capacity for deep reasoning is crucial for success on difficult tasks: on AIME24 (QwQ-32B), FlowSteer achieves **76.7% accuracy**, significantly outperforming AlphaOne’s **53.3%**.

---

> > ### Comment · Reviewer_iiHA · 2026-01-05
> >
> > Dear authors,
> >
> > Thank you for these clarifications; I understand the trade-off now between the reasoning path length and accuracy and agree with you that FlowSteer improves here on AlphaOne. All of my concerns have been addressed. I also appreciate the revisions you have made to the manuscript in response to reviewer feedback; I think those have further strengthened the paper.

---

### Review · Reviewer_hX1Q · 2025-12-12

**Summary Of Contributions:**

This paper proposes a steering method based on non-linear mapping. The authors present novel contributions regarding data normalization, training objectives, and the steering mechanism itself, with rigorous mathematical reasoning. Experiments demonstrate that this approach can mitigate redundant thinking steps, simplify and accelerate the inference process, and improve overall performance.

However, several claims and conclusions in the paper appear to lack sufficient empirical backing. As I am not deeply familiar with this specific topic, I cannot definitively judge whether these statements align with the current consensus in the field. Therefore, while the theoretical analysis is detailed, I have reservations about the reliability of certain statements without further experimental validation or consensus from the broader literature. I have outlined my specific concerns below. If the reviewers can provide supplementary results or citations to substantiate these points, I would consider raising my review result.

Strengths:

1.	The proposed method effectively addresses the limitations of existing steering methods that rely solely on linear mapping for hidden layers. By training a non-linear steering MLP, the paper improves the model's mapping capabilities.

2.	The method is theoretically sound, with rigorous derivations and proofs provided for each component of the framework.

3.	The method demonstrates comprehensive performance improvements over baseline models across multiple datasets.

4.	The trained model exhibits strong generalization ability. It proves not only on different datasets within the same domain as training data but also generalizes well to datasets in other domains, indicating a robust improvement in reasoning capabilities.

Weaknesses:

1.	In Table 2, some results of FlowSteer show only marginal improvement over the second-best method (less than 1%). The authors do not specify whether these results represent a single run or the average of multiple runs. Without this information (and preferably standard deviations), it is difficult to assess the statistical significance and robustness of the results.

2.	The baselines include various steering and prompting methods but exclude any SFT based methods. The "concise reasoning" data used to train the steering vector could directly serve as a training objective for SFT. Although the authors discuss the theoretical distinctions between SFT and steering, and highlight potential downsides of SFT like training speed, the absence of a direct empirical comparison makes it impossible to verify if the proposed method actually outperforms a standard SFT approach in terms of inference quality.

3.	The efficiency analysis in Table 4 is limited, comparing FlowSteer only against the Vanilla method. The inference speeds of other baselines are not included, making it unclear how FlowSteer compares to other steering methods in terms of latency.

**Audience:**

Yes

**Audience Explanation:**

The proposed method is explained clearly with theoretical analysis and experimental results.

**Claims And Evidence:**

Yes

**Claims Explanation:**

See my strengths of summary.

**Requested Changes:**

1.	Additional results are needed to better support the claims.

2.	Including the average performance across datasets in Table 2 would make the comparison clearer.

---

> ### Author Response · Authors · 2025-12-29
> **Thank you for your feedback**
>
> Dear Reviewer hX1Q,
>
> Thank you for your thoughtful and constructive comments on our manuscript.
>
> We have submitted our responses to each point you raised. Please let us know if anything remains unclear or if there are additional issues you would like us to address. We would be happy to elaborate. 🤗
>
> Best regards,
>
> The Authors

---

> > ### Comment · Reviewer_hX1Q · 2025-12-29
> >
> > Thank you for your message. I would like to check whether there might be a system or display issue on my side. At the moment, apart from this message below, I am unable to see any detailed responses to my individual review comments in the system. As a result, I cannot verify how my comments have been addressed.

---

> > > ### Author Response · Authors · 2025-12-29
> > > **Thank you; we have re-submitted the responses**
> > >
> > > Thank you for bringing this to our attention. We hypothesize this may indeed be a display issue within OpenReview. We originally submitted our detailed responses two weeks ago; however, because we posted them before all reviews were released, they may not have rendered correctly for you.
> > >
> > > To resolve this, we have just re-submitted our responses to your comments and deleted the previous versions to avoid confusion. Please let us know if the display issue remains.

---

> > > > ### Comment · Reviewer_hX1Q · 2026-01-05
> > > >
> > > > Thank you for your response. Most of my concerns have been addressed.

---

> ### Author Response · Authors · 2025-12-29
> **Response to the Reviewer hX1Q (1/2)**
>
> We thank the reviewer for the constructive comments. We address the concerns as follows:
>
>
> > Q1.1: Are the results in Table 2 from a single run or multiple runs?
>
>
> Consistent with standard practice in recent reasoning model baselines [1, 2, 3, 4], we report results from a single run. However, to ensure the reliability of our results, we use the following evaluation strategies:
>
> 1. Following protocols in SEAL [1] and LogitsPenalty [2], we use `temperature=0` during greedy decoding to minimize generation variance.
> 2. We evaluated across multiple model sizes (1.5B, 7B, 32B) and diverse benchmarks (Math, Coding). The comparison across these varied settings serves as a proxy for robustness.
>
> We have updated Section 4.1 in the revision to explicitly state these evaluation details. The updated parts are highlighted with the blue color for convenience.
>
> [1] Runjin Chen et al., Steerable reasoning calibration of large language models for free, 2025.
>
> [2] Yue Wang et al., Thoughts are all over the place: On the underthinking of o1-like llms, 2025.
>
> [3] Junyu Zhang et al., Alphaone: Reasoning models thinking slow and fast at test time, 2025.
>
> [4] Silei Xu et al., Chain of draft: Thinking faster by writing less, 2025.
>
>
> > Q1.2: Statistical significance compared to baselines?
>
>
> To rigorously assess the significance of our efficiency gains, we performed a statistical test as detailed in Section 4.3. specifically:
>
> * We treat each pair of (model-dataset) setting as a data point. We apply a one-sided Wilcoxon signed-rank test comparing FlowSteer’s token counts against the strongest baseline in each setting.
> * We restrict this comparison to settings where FlowSteer and the strongest baseline have comparable accuracy (absolute difference < 5 percentage points). This is crucial because there is a **trade-off between accuracy and token count**: harder questions typically require longer CoTs, so a method can trivially “win” on tokens by simply failing early.
> * Under this matched-accuracy protocol, the Wilcoxon test yields a one-sided $ p \approx 0.032 $, which is below the 5% significance level.
>
> This confirms that when accuracy is held comparable, FlowSteer uses fewer tokens.
>
>
>
> > Q1.3: Average performance in Table 2?
>
>
> We have added columns for Average Performance to **Table 2** and visualized these aggregated metrics in **Figures 3 and 4** of the revised manuscript. For convenience, we have also copy-pasted the augmented columns here in **Table A.1-A.3**. These additions demonstrate that FlowSteer achieves a competitive accuracy-token trade-off compared to baselines. Detailed observations from the average performance include:
>
> * **Highest mean accuracy**: FlowSteer consistently achieves the highest average accuracy across all three model sizes.
> * **Competitive accuracy-token count trade-off**: As visualized in the scatter plots, FlowSteer occupies the "top-left" (high accuracy, low tokens) or "top" (highest accuracy) frontier. No other method simultaneously achieves higher accuracy with fewer tokens than FlowSteer. On R1-7B, FlowSteer achieves both the highest accuracy and the lowest token usage.
>
>
> Table A.1: Average performance on R1-1.5B
> | Method        | Avg Acc | Avg T | Avg T@C |
> | ------------- | ------- | ----- | ------- |
> | Vanilla       | 44.2    | 6936  | 2263    |
> | s1*           | 46.4    | 9188  | 7005    |
> | CoD           | 48.1    | 6625  | 2551    |
> | LogitsPenalty | 46.3    | 6590  | 3309    |
> | AlphaOne      | 52.8    | 4329  | 3764    |
> | SEAL          | 56.5    | 5341  | 2633    |
> | FlowSteer     | **58.0**    | 5083  | 2285    |
>
> Table A.2: Average performance on R1-7B
> | Method        | Avg Acc | Avg T | Avg T@C |
> | ------------- | ------- | ----- | ------- |
> | Vanilla       | 68.5    | 5327  | 3221    |
> | s1*           | 66.2    | 6531  | 4466    |
> | CoD           | 68.3    | 4942  | 2830    |
> | LogitsPenalty | 67.4    | 5118  | 2923    |
> | AlphaOne      | 71.2    | 4523  | 3938    |
> | SEAL          | 73.1    | 4605  | 2847    |
> | FlowSteer     | **74.5**    | 4181  | 2563    |
>
> Table A.3: Average performance on QwQ-32B
> | Method        | Avg Acc | Avg T | Avg T@C |
> | ------------- | ------- | ----- | ------- |
> | Vanilla       | 81.4    | 5446  | 4429    |
> | s1*           | 83.8    | 5780  | 4933    |
> | CoD           | 84.2    | 4943  | 4360    |
> | LogitsPenalty | 82.6    | 5161  | 4387    |
> | AlphaOne      | 80.1    | 4079  | 3634    |
> | SEAL          | 82.4    | 5072  | 4154    |
> | FlowSteer     | **87.1**    | 4922  | 4215    |

---

> ### Author Response · Authors · 2025-12-29
> **Response to Reviewer hX1Q (2/2)**
>
> > Q2: Comparison against SFT using the “concise reasoning” data used to train the steering vector?
>
>
> We appreciate this suggestion as it helps isolate the source of our gains. We follow the reviewer’s suggestion and SFT R1-1.5B model for 3 epochs on the same "concise reasoning" data derived from MATH/train used to train the steering vector. The results are presented in **Table B**.
>
> * SFT vs. Vanilla: SFT does improve over the vanilla model (Avg Acc: 51.9% vs. 44.2%) and reduces token usage, confirming that the data itself contains useful signal.
> * FlowSteer vs. SFT: However, FlowSteer outperforms the SFT model, achieving a **+6.1% higher average accuracy** (58.0% vs. 51.9%) while using fewer tokens (5083 vs. 5777).
>
> Thus, under this controlled comparison using identical training data, FlowSteer achieves higher accuracy and lower token usage than standard SFT. We emphasize that this does NOT imply SFT is ineffective: SFT remains a promising direction, especially with better curated or synthesized data and at larger training scales. Rather, SFT and inference-time steering are orthogonal approaches to improving reasoning efficiency, and combining them remains a natural direction for future work.
>
>
> ### Table B: Comparison between FlowSteer and SFT using the same training data.
> | Method | MATH500 Acc. ↑ | MATH500 T. ↓ | GSM8K Acc. ↑ | GSM8K T. ↓ | AIME24 Acc. ↑ | AIME24 T. ↓ | AMC23 Acc. ↑ | AMC23 T. ↓ | LiveCodeBench Acc. ↑ | LiveCodeBench T. ↓ | Avg Acc. ↑ | Avg T. ↓ |
> | --- | --- | --- | --- | --- | --- | --- | --- | --- | --- | --- | --- | --- |
> | Vanilla | 66.6 | 4785 | 73.8 | 2072 | 6.7 | 13807 | 52.5 | 6269 | 21.2 | 7749 | 44.2 | 6936 |
> | SFT | 75.6 | 3507 | **80.2** | 1494 | 16.6 | 11705 | 65.0 | 4536 | 22.2 | 7642 | 51.9 | 5777 |
> | FlowSteer | **79.6** | **2915** | 80.1 | **961** | **33.3** | **10211** | **72.5** | **4111** | **24.5** | **7217** | **58.0** | **5083** |
>
>
> > Q3: Comprehensive accuracy-latency comparison with other baselines?
>
> Thank you for the suggestion. We have revised Section 4.5 and Table 4 to include a comprehensive comparison of average accuracy, TPS, latency, and parameter overhead of all baselines. The extended analysis yields the following key findings:
>
>
> 1. **FlowSteer consistently outperforms the Vanilla baseline**: Across all model sizes (1.5B, 7B, 32B), FlowSteer achieves lower latency than the Vanilla LRM while delivering consistently higher accuracy.
> 2. **Pareto-optimal trade-off on R1-7B and R1-1.5B**: On R1-7B, FlowSteer achieves the lowest latency among all methods ($113.3$s) while attaining the highest accuracy ($74.5\%$). On R1-1.5B, it ranks as the second-fastest method ($133.8$s), surpassed only by AlphaOne ($111.6$s), yet it significantly outperforms AlphaOne in accuracy ($58.0\%$ vs. $52.8\%$).
> 3. **Significant accuracy gain at comparable latency (QwQ-32B)**: While some baseline methods (AlphaOne, CoD) achieve lower latency on the 32B model, they do so at the cost of performance degradation. For example, AlphaOne is faster ($148.3$s) but drops accuracy to $80.1\%$, and CoD ($178.4$s) reaches only $84.2\%$. In contrast, FlowSteer achieves a dominant **87.1% accuracy** while maintaining a latency ($190.0$s) that is still lower than the Vanilla model ($196.6$s).
> 4. **Token reduction as a fundamental algorithmic efficiency**: We emphasize that efficiency is determined by both token counts and TPS/latency. Of these two factors, token reduction represents a fundamental algorithmic efficiency (akin to accuracy improvement), whereas TPS is an implementation-dependent and optimizable metric. While there is a minor drop in TPS compared to the vanilla models due to the flow network, a limitation we explicitly discuss in Section 6, we note that the TPS of FlowSteer may be further improved by employing advanced ODE solvers [5] or faster Flow Matching models [6, 7, 8] in the future work.
>
> Finally, in this work, our primary focus is to compress CoT length to achieve a better **accuracy-token trade-off**.
>
> [5] Cheng Lu et al., Dpm-solver++: Fast solver for guided sampling of diffusion probabilistic models, 2025.
>
> [6] Zhengyang Geng et al., Mean flows for one-step generative modeling, 2025.
>
> [7] Cheng Lu et al., Simplifying, stabilizing and scaling continuous-time consistency models, 2025.
>
> [8] Kevin Frans et. al., One step diffusion via shortcut models, 2025.

---

### Review · Reviewer_p3hQ · 2025-12-22

**Summary Of Contributions:**

The paper contributes FlowSteer, a nonlinear steering framework that leverages Flow Matching to achieve distributional alignment between verbose and concise reasoning, while introducing robust training strategies and a probabilistic guidance mechanism to resolve stability challenges and improve the accuracy-efficiency trade-off in Large Reasoning Models.

Strengths and Weaknesses:

This paper presents an interesting and original idea, and the overall presentation is clear and well organized. The integration of Flow Matching into the proposed framework is appealing, and the empirical results suggest that the approach is effective in practice. However, the paper lacks strong theoretical guarantees. While some theoretical intuition is provided to explain why the method works, these insights remain informal and would benefit from more rigorous justification.

In addition, the method appears to rely on several hyperparameters. In practical applications, tuning such parameters often requires extensive exploration, which may increase computational cost and limit the method’s practicality. It is therefore unclear whether the approach can be considered cost-efficient or robust across different settings.

**Additional Comments:**

NA

**Audience:**

Yes

**Audience Explanation:**

Reasoning in LLMs is an inherently interesting research topic. Integrating Flow Matching into LLM reasoning tasks yields a new promising research direction.

**Claims And Evidence:**

Yes

**Claims Explanation:**

1. The authors clearly convey the core concepts and procedural steps of their approach, making the general idea easy to understand.

2. The study is supported by comprehensive experiments conducted across 3 LLMs and 5 diverse datasets, demonstrating robust evaluation.

3. While FlowSteer does not show significant improvements in every single task or across all LLMs, it consistently performs either better than or close to state-of-the-art baselines like SEAL. Notably, FlowSteer often achieves comparable accuracy with greater token efficiency.

**Requested Changes:**

Questions:

1. Section 3.1 (Page 5): Loss Function Choice
   As I understand it, the final loss used in practice is the Huber loss rather than the mean squared error (MSE). However, Equation (4) defines the MSE in expectation, whereas the Huber loss is presented without an expectation. Is the Huber loss also optimized in expectation, or is there a different interpretation?

2. Section 3.2 (Page 5): Use of the High-Dimensional Hypothesis
   The paper states: *“This provided evidence … original high-dimensional space.”* It is unclear where exactly this hypothesis is used prior to this statement. Could the authors explicitly indicate where this assumption is introduced and how it is leveraged in the methodology or analysis?

3. Section 3.2 (Page 6): Parameter-Free Guidance
   The paper claims to derive a *parameter-free guidance*, yet the method requires estimating the mean and variance. This seems contradictory. Could the authors clarify what is meant by “parameter-free” in this context? Does it mean that no additional trainable parameters are introduced, rather than no parameters at all?

4. Appendix A (Page 18): Sensitivity to η
   The reported values of η lie in a very narrow range (minimum 0.85, maximum 1.15). Does the method fail or become unstable for smaller or larger values of η? This appears to be a tight range, and some discussion of robustness or collapse behaviour would be helpful.

Writing and Presentation Suggestions:

5. Page 2:
   The term *“Principle”* should be corrected to *“Principal”* when referring to Principal Component Analysis (PCA).

6. Terminology Consistency:
   The method name appears inconsistently as *Flowsteer*, *FlowSteer*, and *FLOWSTEER* (e.g., Figure 1). Please use a single, consistent naming convention throughout the paper.

7. Text Coloring:
   The use of blue-colored text (e.g., Pages 7–11) is unnecessary and distracting. Standard black text would improve readability and professionalism.

8. References:
   Several references are cited only as arXiv preprints. For works that have been published in conferences or journals, please cite the official publication venues instead of the arXiv versions. This is important for maintaining citation quality and formality during the review process.

---

> ### Author Response · Authors · 2025-12-29
> **Response to Reviewer p3hQ (1/2)**
>
> We thank the reviewer for the valuable comments and suggestions. We have carefully considered each point and provide our responses below:
>
> > W1: While some theoretical intuition is provided to explain why the method works, these insights remain informal and would benefit from more rigorous justification.
>
> We agree with the reviewer that formal theoretical guarantees would further strengthen the work. We acknowledge that our contribution focuses on the proposed **algorithmic framework** and its empirical validation rather than deriving new theories.
>
> Importantly, FlowSteer is a principled adaptation of the established theory of Flow Matching (FM) to large reasoning models. The theoretical properties of FM, especially its capability for target distribution preservation, systematically guided the design of our approach. In this work, we prioritize establishing the practical efficacy of our algorithmic framework. We rely on our extensive empirical validation to demonstrate that these theoretically grounded objectives translate effectively to steering LRMs. We agree with the reviewer that complementing this empirical evidence with formal analysis would further solidify the foundations of non-linear steering.
>
> > W2:  The method appears to rely on several hyperparameters…It is therefore unclear whether the approach can be considered cost-efficient or robust across different settings.
>
> and
>
> > Q4: Does the method fail or become unstable for smaller or larger values of η? This appears to be a tight range, and some discussion of robustness or collapse behaviour would be helpful.
>
>
> We respectfully clarify that FlowSteer introduces a new hyperparameter, the probabilistic guidance strength $\eta$. This parameter is analogous to the standard classifier-free guidance scale widely utilized in the diffusion and flow matching literature. The other hyperparameters, such as the steering layer, are adopted directly from SEAL.
>
> To address concerns regarding robustness of the choice of $\eta$, we have updated **Appendix D.2** and **Figure 5**, expanding the ablation range of $\eta$ from $\[0.85, 1.15\]$ to $\[0.1, 2.0\]$. The results in Figure 5 confirm the robustness of our method:
>
> * significant improvements in both accuracy and token reduction persist at $\eta=0.1$ and $\eta=1.5$, indicating a broad effective range.
>
> * At the extreme of $\eta=2.0$, while token usage is still reduced effectively, the accuracy gains diminish towards the vanilla baseline.
>
> * Notably, $\eta=1.0$ emerged as the optimal choice in 10 out of 15 experimental settings (**Table 2**), establishing it as a reliable default value for practical applications.
>
> Mechanistically, we analyze the behavior in the extreme regimes as follows.
>
> * In the low-guidance regime ($η=0.1$), the force provided by the probabilistic guidance is too small; while it helps prevent stagnation in "low-velocity zones," it is less effective at steering the trajectory than the optimal setting ($η \approx 1.0$).
>
> * Conversely, in the high-guidance regime ($\eta=2.0$), the guidance term dominates the ODE dynamics (Equation (8)). This suppresses the flow model, which is responsible for capturing the fine-grained structure of the target representation distribution. Consequently, the trajectory relies too heavily on the coarser guidance approximation, sacrificing the detailed geometric information necessary for better reasoning performance.
>
> This discussion has been incorporated into the revised Appendix D.2.
>
> > Q1: Is the Huber loss also optimized in expectation, or is there a different interpretation?
>
> The Huber loss is optimized in expectation. Specifically, the Huber loss replaces the MSE function within the expectation in Equation (4). In Equation (6), we omitted the expectation operator to improve readability and to focus on the formulation of the Huber loss itself. We have added one sentence after Equation (6) to make this distinction explicit.

---

> ### Author Response · Authors · 2025-12-29
> **Response to Reviewer p3hQ (2/2)**
>
> > Q2: The paper states: “This provided evidence … original high-dimensional space.” It is unclear where exactly this hypothesis is used prior to this statement. Could the authors explicitly indicate where this assumption is introduced and how it is leveraged in the methodology or analysis?
>
> The hypothesis refers to the significant **spatial proximity** between the source and target distributions in the representation space. In Section 3.2, this assumption serves as the logical premise for our analysis of "low-velocity zones." Specifically, prior studies e.g. Section 3 in the SEAL paper [1] suggests that these distributions remain spatially close. This proximity is the specific condition that triggers the "low-velocity" issue: it naturally generates training pairs $(x_0​, x_1​)$ with small-magnitude velocity targets $(x_1​ − x_0​)$, forcing the model to learn a stagnant field in these regions. We have revised Section 3.2 to explicitly clarify this point and remove ambiguity.
>
> [1] Runjin Chen, Zhenyu Zhang, Junyuan Hong, Souvik Kundu, and Zhangyang Wang. Seal: Steerable reasoning calibration of large language models for free, COLM, 2025.
>
> > Q3:  Could the authors clarify what is meant by “parameter-free” in this context “parameter-free guidance”? Does it mean that no additional trainable parameters are introduced, rather than no parameters at all?
>
> The "parameter-free" in this context specifically means that no additional trainable parameters are introduced. We use this term to distinguish our probabilistic guidance from conventional classifier guidance [2], which typically necessitates training an auxiliary classifier model. In contrast, our method derives guidance solely from the statistics (mean and covariance) of the hidden states, requiring no new weights. We have changed “parameter-free guidance” to “training-free guidance” in the revised Section 3.2.
>
> [2] Prafulla Dhariwal and Alexander Nichol. Diffusion models beat gans on image synthesis. Advances in neural information processing systems, 34:8780–8794, 2021.
>
> ---
>
> ### Responses to the Suggestions:
>
> > S5: Page 2: The term “Principle” should be corrected to “Principal”
>
> We thank the reviewer for the suggestion, and we have updated the term in the revised manuscript.
>
> > S6: Please use a single, consistent naming convention throughout the paper.
>
> We thank the reviewer for raising this point, and we have updated the manuscript to use `FlowSteer` instead of `\textsc{FlowSteer}` throughout the paper.
>
> > S7: Text Coloring: The use of blue-colored text (e.g., Pages 7–11) is unnecessary
>
> We have highlighted our revisions in blue to indicate the changes made in response to other reviewers' suggestions, helping all reviewers to locate these updates easily. In the final manuscript, this blue highlighting will be removed to maintain consistent document formatting.
>
> > S8: For works that have been published in conferences or journals, please cite the official publication venues instead of the arXiv versions.
>
> We thank the reviewer for the suggestion. We have updated the citations and ensured that the latest and official versions are cited where applicable.

---

> ### Author Response · Authors · 2025-12-29
> **Thank you for your feedback**
>
> Dear Reviewer p3hQ,
>
> Thank you for your thoughtful and constructive comments on our manuscript.
>
> We have submitted our responses to each point you raised. Please let us know if anything remains unclear or if there are additional issues you would like us to address. We would be happy to elaborate. 🤗
>
> Best regards,
>
> The Authors

---

> > ### Comment · Reviewer_p3hQ · 2025-12-31
> >
> > I thank the authors for their detailed and thoughtful response. Most of my questions have been satisfactorily addressed, and the revised manuscript shows clear improvements in both quality and clarity. Please see below some further concerns:
> >
> > Q1 (Huber loss): It would be helpful to explicitly state that the optimization is performed over the expectation of the loss, to avoid any potential ambiguity.
> >
> > Q8 (References and consistency): While several writing issues have been addressed, some reference-related and consistency issues remain. For instance, *“Do not think that much for 2 + 3 = ? On the overthinking of o1-like LLMs”* has been accepted to ICML 2025. Please replace the corresponding arXiv citation with the official conference version, and more generally, cite the final publication venues instead of arXiv preprints wherever possible.
> >
> > Addressing these remaining points would further improve the readability and professionalism of the paper.

---

> > > ### Author Response · Authors · 2025-12-31
> > > **Further updates on manuscript**
> > >
> > > We are glad that the main concerns have been addressed and appreciate the final presentation suggestions. We have incorporated these suggestions into the revised manuscript as follows:
> > >
> > > * **Q1 (Huber loss):** We have clarified the text following Eq. (6) to explicitly state the optimization objective:
> > >     > The total loss is obtained by averaging across all $d$ dimensions. This loss replaces the MSE loss in Eq. (4), and the final optimization objective is the expectation of this total Huber loss over time $t \sim \mathcal{U}[0,1]$ and data pairs $ (x_0, x_1) \sim \pi_{0,1} $.
> > >
> > > * **Q8 (References):** We have standardized the bibliography to cite final publication venues instead of arXiv preprints wherever possible.
> > >
> > > Thank you again for your time and for the constructive feedback that helped improve our work.

---

> > > > ### Comment · Reviewer_p3hQ · 2026-01-04
> > > >
> > > > Thanks a lot, my questions and concerns have been addressed.

---

### Decision · Action_Editor_yWAG · 2026-01-19

**Recommendation:** Accept as is

**Audience:**

Yes

**Audience Explanation:**

The paper tackles an important problem, that is improving the accuracy-efficiency tradeoff of Large Reasoning Models through latent steering strategies. This work would be of interest to the individuals working on LLMs and generative models.

**Claims And Evidence:**

Yes

**Claims Explanation:**

The paper contributes with a latent steering strategy for Large Reasoning Models achieving a higher accuracy-efficiency tradeoff in math and code domains. The steering method leverages a Flow matching model to translate states from longer to shorter reasoning paths. Existing latent steering strategies are based on linear translation. This work contributes to extending the family of steering strategies to the nonlinear case.

All reviewers appreciated the solid experimental methodology and the simplicity of the idea. The paper makes an interesting empirical contribution leaving room for future theoretical work.